# Mapping the functional landscape of the receptor binding domain of T7 bacteriophage by deep mutational scanning

Phil Huss[1,2], Anthony Meger[1], Megan Leander[1], Kyle Nishikawa[1], Srivatsan Raman[1,2,3]*

[1]Department of Biochemistry, University of Wisconsin-Madison, Madison, United States; [2]Department of Bacteriology, University of Wisconsin-Madison, Madison, United States; [3]Department of Chemical and Biological Engineering, University of Wisconsin-Madison, Madison, United States

**Abstract** The interaction between a bacteriophage and its host is mediated by the phage's receptor binding protein (RBP). Despite its fundamental role in governing phage activity and host range, molecular rules of RBP function remain a mystery. Here, we systematically dissect the functional role of every residue in the tip domain of T7 phage RBP (1660 variants) by developing a high-throughput, locus-specific, phage engineering method. This rich dataset allowed us to cross compare functional profiles across hosts to precisely identify regions of functional importance, many of which were previously unknown. Substitution patterns showed host-specific differences in position and physicochemical properties of mutations, revealing molecular adaptation to individual hosts. We discovered gain-of-function variants against resistant hosts and host-constricting variants that eliminated certain hosts. To demonstrate therapeutic utility, we engineered highly active T7 variants against a urinary tract pathogen. Our approach presents a generalized framework for characterizing sequence–function relationships in many phage–bacterial systems.

**\*For correspondence:**
sraman4@wisc.edu

## Introduction

Bacteriophages (or 'phages') shape microbial ecosystems by infecting and killing targeted bacterial species. As a result, they are promising tools for treatment of antibiotic-resistant bacterial infections and microbiome manipulation (*Canfield and Duerkop, 2020*; *Chen et al., 2014*; *Clokie et al., 2011*; *Dedrick et al., 2019*; *Kilcher and Loessner, 2019*; *Kutter et al., 2015*; *Mizuno et al., 2020*; *Sausset et al., 2020*; *Schooley et al., 2017*; *Shen et al., 2015*; *Shkoporov and Hill, 2019*; *De Sordi et al., 2019*). Interaction of phages with their bacterial receptors is a key determinant of their host range and virulence (*Bertozzi Silva et al., 2016*; *de Jonge et al., 2019*; *Rousset et al., 2018*). This interaction is primarily mediated by the receptor binding proteins (RBPs) of the phage (*Nobrega et al., 2018*). RBPs enable phages to adsorb to diverse cell surface molecules, including proteins, polysaccharides, lipopolysaccharides (LPS), and carbohydrate-binding moieties. Phages exhibit high functional plasticity through genetic alterations to RBPs and by natural and laboratory-guided evolution, which can modulate activity and host range to different hosts and environments (*Ando et al., 2015*; *Chen et al., 2017*; *Dedrick et al., 2019*; *Dunne et al., 2019*; *Garcia et al., 2003*; *Gebhart et al., 2017*; *Holtzman et al., 2020*; *Lin et al., 2012*; *Meyer et al., 2012*; *Yehl et al., 2019*; *Yosef et al., 2017*). In essence, survivability of a phage is intimately linked to the adaptability of its RBP. The challenge now is to understand the molecular code of RBPs in sufficient

**eLife digest** Bacteria can cause diseases, but they also battle their own microscopic enemies: a group of viruses known as bacteriophages. For instance, the T7 bacteriophage preys on various strains of *Escherichia coli*, a type of bacteria often found in the human gut. While many *E. coli* strains are inoffensive or even beneficial to human health, some can be deadly. Finding a way to kill harmful strains while sparing the helpful ones would be a helpful addition to the medicine toolkit.

Bacteriophages identify and interact with their specific target through a structure known as the receptor binding protein, or RBP. However, it is still unclear exactly how RBP helps the viruses recognize which type of bacteria to infect. Here, Huss et al. set to map out and modify this structure in T7 bacteriophage so the virus is more efficient and specific about which strain of *E. coli* it kills.

First, the role of each building block in the tip of RBP was meticulously dissected; this generated the knowledge required to genetically engineer a large number of different T7 bacteriophages, each with a slightly variation in their RBP. These viruses were then exposed to various strains of bacteria. Monitoring the bacteriophages that survived and multiplied the most after infecting different strains of *E. coli* revealed which RBP building blocks are important for efficiency and specificity. This was then confirmed by engineering highly active T7 bacteriophage variants against an *E. coli* strain that causes urinary tract infections.

These findings demonstrate that even small changes to the bacteriophages can make a big difference to their ability to infect their preys. The approaches developed by Huss et al. help to understand exactly how the RBP allows a virus to infect a specific type of bacteria; this could one day pave the way for new therapies that harness those viruses to fight increasingly resistant bacterial infections.

depth to enable predictable manipulation of host range and virulence. We sought to do so by combining deep mutational scanning (DMS) of the RBP with powerful selections on multiple hosts.

Although RBPs remain the focus of many mechanistic, structural, and evolutionary studies and are a prime target for engineering, we currently lack a systematic and comprehensive understanding of how RBP mutations influence phage activity and host range. Though insightful, directed evolution enriches only a small group of 'winners', which makes it difficult to glean a comprehensive mutational landscape of the RBP (*Holtzman et al., 2020*). Random mutagenesis-based screens generate multi-mutant variants whose individual effects cannot be easily deconvolved (*Dunne et al., 2019*; *Yehl et al., 2019*). Other approaches including swapping homologous RBPs lead to gain of function; however, the underlying molecular determinants of function can be difficult to explain (*Ando et al., 2015*; *Chen et al., 2017*; *Gebhart et al., 2017*; *Yosef et al., 2017*). In summary, despite the extraordinary functional potential of phage RBPs, how systematic changes to their sequence shape the overall functional landscape of a phage remains unknown.

Here, we carried out DMS, a high-throughput experimental technique, of the tip domain of the T7 phage RBP (tail fiber) to uncover molecular determinants of activity and host range. The tip domain is the distal region of the tail fiber that makes primary contact with the host receptor (*González-García et al., 2015*; *Molineux, 2001*; *Qimron et al., 2006*). We developed ORACLE (**O**ptimized **R**ecombination, **Ac**cumulation, and **L**ibrary **E**xpression), a high-throughput, locus-specific, phage genome engineering method to create a large, unbiased library of phage variants at a targeted gene locus. Using ORACLE, we systematically and comprehensively mutated the tip domain by making all single amino acid substitutions at every site (1660 variants) and quantified the functional role of all variants on multiple bacterial hosts. We generated high-resolution functional maps delineating regions concentrated with function-enhancing substitutions and host-specific substitutional patterns, many of which were previously unknown. We discovered T7 variants with far greater virulence than wildtype T7, demonstrating that even those natural phages that are well adapted to a host can be engineered for higher efficacy.

However, many variants highly adapted to one host performed poorly on others, underscoring a tradeoff between activity and host range. This functional screening highlights ideal regions of the tip domain for engineering host range. Furthermore, we demonstrated the functional potential of RBPs by discovering gain-of-function variants against resistant hosts and host-constriction variants that

selectively eliminate specific hosts. To demonstrate the therapeutic value of ORACLE, we engineered T7 variants that avert emergence of spontaneous resistance in pathogenic *Escherichia coli* causing urinary tract infections (UTIs).

Our study explains the molecular drivers of adaptability of the tip domain and identifies key functional regions determining activity and host range. ORACLE provides a generalized framework to describe sequence–function relationships in phages to elucidate the molecular basis of phages, the most abundant life form on earth.

## Results

### Creating an unbiased library of phage variants using ORACLE

ORACLE is a high-throughput precision phage genome engineering technology designed to create a large, unbiased library of phage variants to investigate sequence–function relationships in phages. ORACLE overcomes three major hurdles. First, phage variants are created during the natural infection cycle of the phage, which eliminates a common bottleneck from transforming DNA libraries. By recombining a donor cassette containing prespecified variants to a targeted site on the phage genome, ORACLE allows sequence programmability and generalizability to phages with transformable bacterial hosts capable of maintaining a plasmid library. Second, ORACLE minimizes library bias that can rapidly arise due to fitness advantage or deficiency of any variant on the propagating host that may then be amplified due to exponential phage growth. Minimizing bias is critical because variants that perform poorly on a propagating host but well on targeted hosts may disappear during propagation. Third, ORACLE prevents extreme abundance of wildtype over variants, which allows for resolving and scoring even small functional differences between variants. The development of this technology was necessary to overcome challenges with existing engineering approaches for creating a large, unbiased phage library. Direct transformation of phage libraries, while ideal for creating one or small groups of synthetic phages, will not work because phage genomes are typically too large for library transformation (*Ando et al., 2015*; *Kilcher et al., 2018*; *Marinelli et al., 2008*; *Marinelli et al., 2019*). Homologous recombination has low, variable recombination rates and high levels of wildtype phage are retained, which mask library members (*Pires et al., 2016*; *Yehl et al., 2019*). Libraries of lysogenic phages could potentially be made using conventional bacterial genome engineering tools as the phage integrates into the host genome. However, this approach is not applicable to obligate lytic phages. Our desire to develop ORACLE for obligate lytic phages is motivated by their mandated use for phage therapy. Any phage, including lysogenic phages, with a sequenced genome and a transformable host that can maintain a plasmid library should be amenable to ORACLE.

ORACLE is carried out in four steps: (a) making acceptor phage, (b) inserting gene variants through recombination, (c) accumulating recombined phages, and (d) expressing the library for selection (*Figure 1A*). An 'acceptor phage' is a synthetic phage genome where the gene of interest (i.e., tail fiber) is replaced with a fixed sequence flanked by Cre recombinase sites to serve as a landing site for inserting variants (*Figure 1—figure supplement 1*). We created T7 acceptor phages by assembling PCR fragments of the phage genome in yeast (*Ando et al., 2015*; *Jaschke et al., 2012*) (see Materials and methods). T7 acceptor phages lacking a wildtype tail fiber gene cannot plaque on *E. coli* and do not spontaneously reacquire the tail fiber during propagation (*Figure 1B, Figure 1—figure supplement 2A*). Furthermore, the T7 acceptor phages have no plaquing deficiency relative to wildtype when the tail fiber gene is provided from a helper plasmid (*Figure 1—figure supplement 2A*). Thus, the tail fiber gene is decoupled from the rest of the phage genome for interrogation of function. Next, phage variants are generated within the host during the infection cycle by **O**ptimized **R**ecombination by inserting tail fiber variants from a donor plasmid into the landing sites in the acceptor phage using site-specific recombination. To minimize biasing of variants during propagation, a helper plasmid constitutively provides the wildtype tail fiber in trans such that all progeny phages can amplify comparably regardless of the fitness benefit or deficient of any variant. At this stage, we typically have approximately 1 recombined phage among 1000 acceptor phages (*Figure 1C*). To enrich recombined phages in this pool, we passage all progeny phages on *E. coli* expressing Cas9 and a gRNA targeting the fixed sequence flanked by recombinase sites we introduced into the acceptor phage. The helper plasmid is retained during this stage to continue

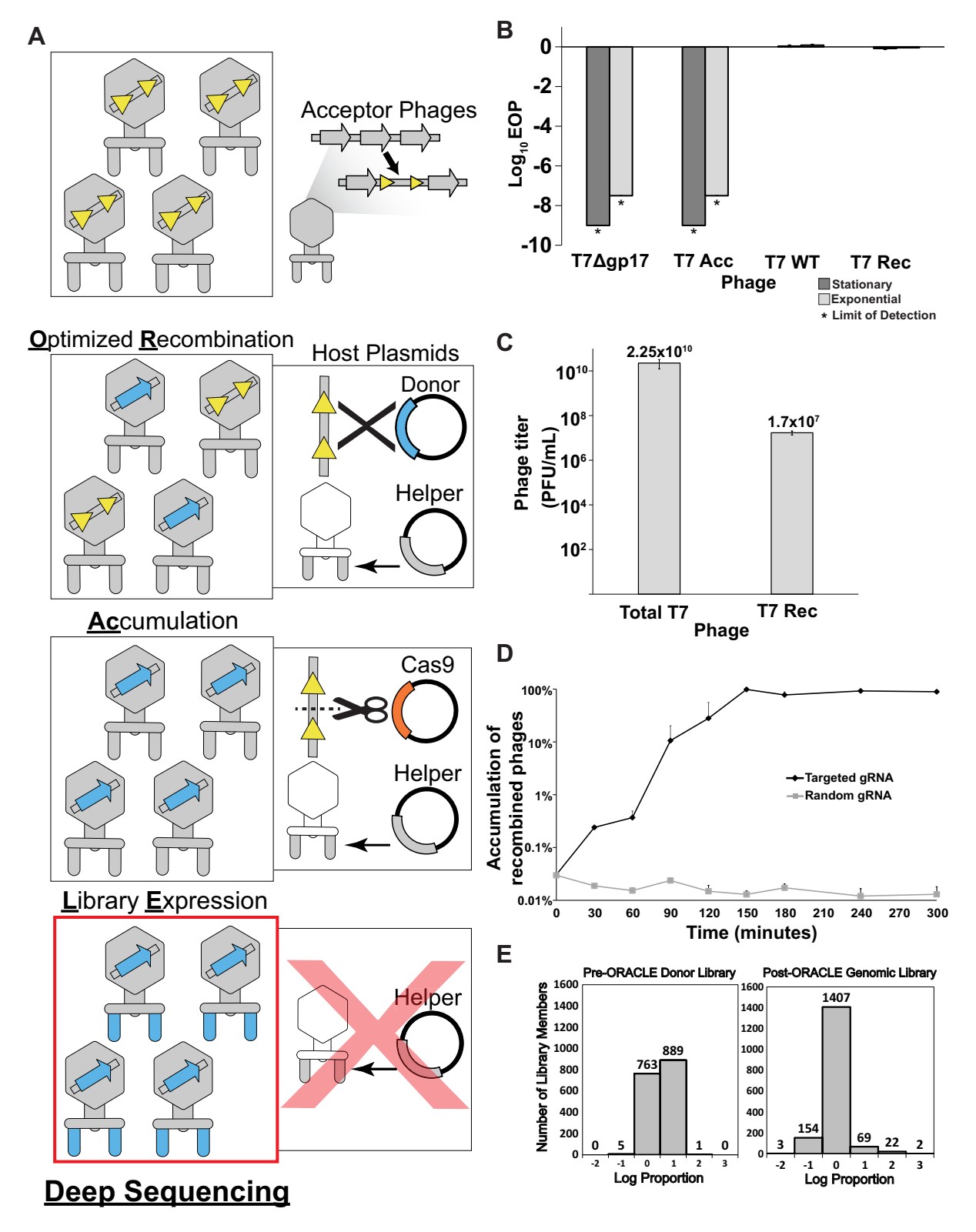

**Figure 1.** Optimized Recombination, Accumulation, and Library Expression (ORACLE) workflow for creating phage variant libraries. (**A**) Schematic illustration of the four steps of ORACLE: creation of acceptor phage, inserting gene variants (Optimized Recombination), enriching recombined phages (Accumulation) and expressing library for selection (Library Expression). Color notations are as follows: yellow triangles – Cre recombinase sites, blue colored segments – gene variants, orange colored segment – Cas9, grey colored segments – wild type phage parts including the wildtype tail fiber

*Figure 1 continued on next page*

*Figure 1 continued*

from the helper plasmid (**B**) Ability of different versions of T7 to infect *E. coli* 10G in stationary (dark gray bar) and exponential (light gray bar) phases by Efficiency of Plating (EOP) using exponential 10G with *gp17* tail fiber helper plasmid as reference host. T7 without tail fiber (T7Δgp17) and T7 Acceptor phages (T7 Acc) cannot visibly plaque, but wildtype T7 (T7 WT), and T7 with *gp17* recombined into the acceptor locus (T7 Rec) plaque efficiently. (**C**) Concentration of total (Total T7) and recombined (T7 Rec) phages after a single passage on host containing Cre recombinase system. Recombination rate is estimated to be ~$7.19 \times 10^{-4}$. (**D**) Percentage of recombined phages in total phages when using gRNA targeting fixed sequence at acceptor site T7 Acc (Targeted) or randomized gRNA (Random). (**E**) Histogram of abundance of variants in the input plasmid library (left) and on the phage genome after ORACLE (right) binned using log proportion centered on equal representation. All data represented as mean ± SD of biological triplicate.

The online version of this article includes the following source data and figure supplement(s) for figure 1:

**Source data 1.** Deep sequencing summary for phage variant expression library with and without DNAse treatment after **O**ptimized **R**ecombination, **Ac**cumulation, and **L**ibrary **E**xpression (ORACLE).
**Source data 2.** Percentage distribution of each variant in the expression library.
**Figure supplement 1.** Sequence rearrangements before and after recombinase-mediated cassette exchange.
**Figure supplement 2.** Effect of Cas9-gRNA system on acceptor and control phages.
**Figure supplement 3.** Correlation between expression library with and without DNAse treatment.

minimizing bias by providing the wildtype tail fiber in trans. As a result, only unrecombined phages will be inhibited while recombined phages with tail fiber variants are **Ac**cumulated without bias. The Cas9-gRNA system successfully inhibits acceptor phages but has no effect on plaquing of untargeted phages (*Figure 1—figure supplement 2A–D*). Recombined phages were highly enriched by over one thousandfold in the phage population when an optimized gRNA targeting the fixed sequence was used, whereas a randomized control gRNA yielded no enrichment of recombined phages (*Figure 1D, Figure 1—figure supplement 2E, F*). In the final step, phages are propagated on *E. coli* which lack the helper plasmid that previously provided the wildtype tail fiber in trans to prevent bias. In this **L**ibrary **E**xpression, propagation on this host allows for full expression of the library variant – this is the first time during library creation that the variant is fully expressed on the phage particle. We sequenced the distribution of the library of tail fiber variants integrated on the phage genome after ORACLE. We compared this distribution to the distribution of variants on the recombination plasmid library to evaluate how effective ORACLE was at integrating variants and preventing bias during library creation. The post-ORACLE phages were mildly skewed toward more abundant members but remained generally evenly distributed and comparable to the distribution of variants in the input donor plasmid library, retaining 99.8% coverage (*Figure 1E*). Comparison of variant libraries with and without DNAse treatment was well correlated (R = 0.994), indicating no unencapsidated phage genomes influenced library distribution (*Figure 1—figure supplement 3*). In summary, ORACLE is a generalizable tool for creating large, unbiased variant libraries of obligate lytic phages. These phage variants, including those that have a fitness deficiency on the host used to create the library, can all be characterized in a single selection experiment by deep sequencing phage populations before and after selection in a host. Compared to traditional plaque assays, this represents increased throughput by nearly 3–4 orders of magnitude.

## DMS of the tip domain shows phage adaptation at molecular resolution

DMS is a high-throughput experimental technique to characterize sequence–function relationships through large-scale mutagenesis coupled to selection and deep sequencing. The scale and depth of DMS is used to reveal sites critical for activity, host specificity, and stability in a protein. DMS has been employed to study many proteins, including enzymes, transcription factors, signaling domains, and viral surface proteins (*Fowler and Fields, 2014*; *Lee et al., 2018*; *Raman et al., 2014*; *Romero et al., 2015*).

Bacteriophage T7 is a podovirus that infects *E. coli*. T7 has a short non-contractile tail made up of three proteins, including the tail fiber encoded by *gp17*. Each of the six tail fibers is a homotrimer composed of a relatively rigid shaft ending with a β-sandwich tip domain connected by a short loop (*Garcia-Doval and van Raaij, 2012*). The tip domain is likely the very first region of the tail fiber to interact with host LPS and position the phage for successful, irreversible binding with the host (*González-García et al., 2015*; *Molineux, 2001*; *Qimron et al., 2006*). The tip domain is a major determinant of host range and activity and is often naturally exchanged between phages to readily adapt to new hosts (*Fraser et al., 2006*; *Fraser et al., 2007*; *Lin et al., 2012*). Even single amino acid

substitutions to this domain are sufficient to alter host range between *E. coli* strains (*Heineman et al., 2008*). Due to its critical functional role, we chose the tip domain to comprehensively characterize phage activity and host range by DMS.

We generated a library of 1660 single mutation variants of the tip domain, prespecified as chip-based oligonucleotides, where all 19 non-synonymous and 1 nonsense substitution were made at

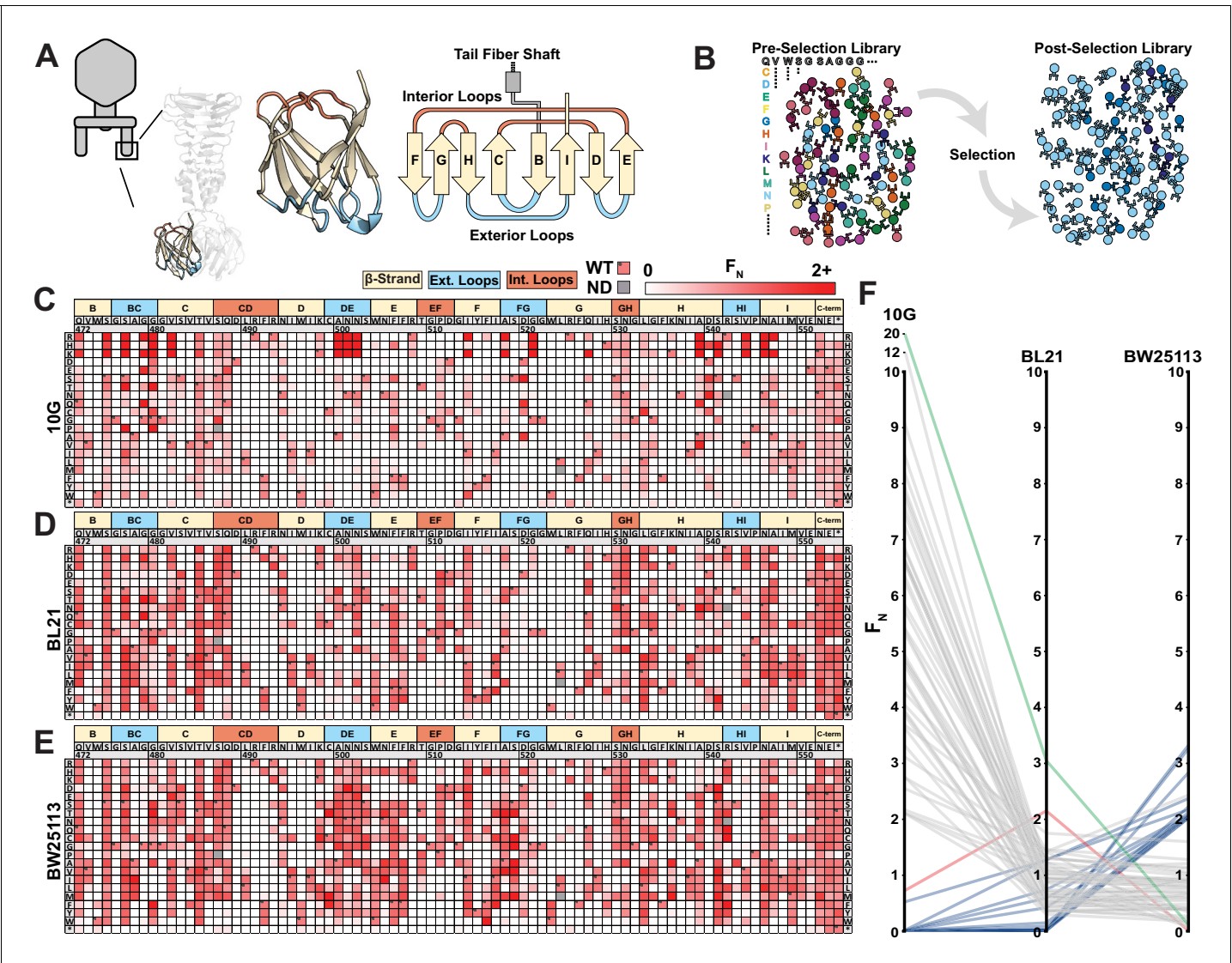

**Figure 2.** Deep mutational scanning of tip domain shows phage adaptation at molecular resolution. (**A**) Crystal structure and secondary structure topology of the tip domain color coded as interior loops (red), β-sheets (beige) and exterior loops (blue) (**B**) Functional analysis of variants by comparing their abundances pre- and post-selection on a host. (**C-E**) Heat maps showing normalized functional scores ($F_N$) of all substitutions (red gradient) and wildtype amino acid ($F_N$=1 and black dot upper left) at every position for *E. coli* 10G (**C**), BL21 (**D**) and BW25113 (**E**). Residue numbering (based on PDB 4A0T), wildtype amino acid and secondary structure topology are shown above left to right, substitutions listed top to bottom. (**F**) Parallel plot showing $F_N$ for enriched ($F_N \geq 2$) variants on 10G, BL21, and BW25113. Coloring indicates enrichment only on 10G (grey), only on BL21 (red), only on BW25113 (blue) enriched on 10G and BL21 (green). Connecting lines indicate $F_N$ of the same variant on other hosts.

The online version of this article includes the following source data and figure supplement(s) for figure 2:

**Source data 1.** Deep sequencing summary for the phage variant library after selection on different hosts.

**Source data 2.** Variant-specific $F_N$ for phage variants after selection on *E. coli* 10G, BL21 ,and BW25113 and physicochemical statistics.

**Figure supplement 1.** Correlation between biological replicates after selection of phage variant library on *E. coli* 10G, BL21, and BW25113.

**Figure supplement 2.** Distribution, enrichment profile, and physicochemical properties of variants after selection on *E. coli* 10G, BL21, and BW25113.

**Figure supplement 3.** Hierarchical clustering of phage variants based on $F_N$ score.

each codon spanning residue positions 472–554 (*Figure 2A*, residue numbering based on PDB 4A0T). Using ORACLE, the library was inserted into T7 to generate variants to be selected and deep sequenced (*Figure 2B*) on three laboratory *E. coli* hosts: B strain derivative BL21, K-12 derivative BW25113, and DH10B derivative 10G. Each variant was given a functional score, F, based on the ratio of their relative abundance before and after selection consisting of an estimated four infection cycles, which was then normalized to wildtype to yield $F_N$, where wildtype $F_N = 1$ (*Figure 2C–E*, see Materials and methods). Selection on each host gave excellent correlation across biological triplicates (*Figure 2—figure supplement 1*). To validate the functional relevance of the screen, we hypothesized that the flexible C-terminal end (residue positions 552–554 and a three-residue extension if the stop codon is substituted) is unlikely to have any structural or host recognition role. As expected, these positions broadly tolerated nearly all substitutions across all three hosts, indicating that the functional scores likely reflect true biological effects (*Figure 2C–E*).

We compared the activities of phage variants across hosts to assess their fitness and evolutionary adaptation to each host. Between the three hosts, T7 variants appeared most and least adapted to BW25113 and 10G, respectively, as evidenced by the fraction of depleted variants ($F_N < 0.1$) after selection on each host (10G: $0.66 \pm 0.03$; BL21: $0.59 \pm 0.01$; and BW25113: $0.51 \pm 0.01$; all significantly different from each other with $p<0.05$) (*Figure 2—figure supplement 2A–C*). Furthermore, wildtype T7 fared relatively poorly on 10G ($F = 0.77 \pm 0.05$), indicating a fitness impediment, but performed significantly better on BL21 ($F = 2.92 \pm 0.2$, $p < 0.01$) and BW25113 ($F = 2.26 \pm 0.1$, $p < 0.01$) (*Figure 2—source data 1*). The fitness impediment gave many more variants competitive advantage, resulting in greater enrichment ($F_N > 2$) over wildtype on 10G (48 variants) compared to BL21 (2 variants) and BW25113 (16 variants) (*Figure 2—figure supplement 2A–C*). In fact, the best performing variants on 10G were 10 times more enriched than wildtype, suggesting substantially higher activity (*Figure 2F, Figure 2—figure supplement 2D*). Examining enriched variants on each host ($F_N > 2$) provides compelling evidence of the tradeoff between activity and host range (*Figure 2F, Figure 2—figure supplement 2E*). The top ranked variants on each host were remarkably distinct from those on other hosts (except G479Q shared between 10G and BL21). Hierarchical clustering of $F_N$ across all three hosts revealed grouping of similar variants that performed better selectively on some hosts but not others (*Figure 2—figure supplement 3*). No variant performed exceptionally well on all hosts ($F_N > 2$, *Figure 2F*); however, 406 variants were tolerated on all three hosts (*Figure 2—source data 2*). Thus, specialization toward a host comes at the cost of sacrificing breadth, mirroring observations made of natural phage populations (*Elena et al., 2009*).

We investigated the global physicochemical properties and topological preferences of substitutions after selection on each host (*Figure 2—figure supplement 2F–H*). On 10G, there was enrichment of larger and more hydrophilic amino acids and depletion of hydrophobic amino acids (all $p < 0.001$, $r > 0.12$), which is visually striking on the heatmap (see R, K, and H substitutions in *Figure 2C*). In contrast, no significant enrichment or depletion was observed on BL21 (*Figure 2—figure supplement 2F–H*). This is consistent with our earlier observation that wildtype T7 is generally well adapted to BL21 since it had the fewest variants outperforming wildtype. We reasoned that since BL21 has been used to propagate T7 it may have already adapted well to this host over time. On BW25113, hydrophobic residues were modestly enriched (all $p < 0.034$, $r > 0.07$) (*Figure 2—figure supplement 2H*), a trend opposite to 10G. This provides a molecular explanation as to why high-scoring substitutions on one host fare poorly on others (*Figure 2F*). We mapped positions of enriched substitutions ($F_N \geq 2$) on each host onto the structure to determine topologically distinct patterns of substitution that may be masked in global comparisons of the entire tip domain (*Figure 2—figure supplement 2E*). These fall predominantly on four exterior loops (BC, DE, FG, and HI), the adjoining region (β-strand H) close to exterior loop HI, and less frequently on the 'side' of the tip domain. This suggests directionality to phage–bacterial interactions and orientational bias of the tip domain with respect to the bacterial surface. Directionality and orientational bias is particularly valuable information since no high-resolution structure of this phage bound to receptor exists.

Several key lessons emerged from these host screens. First, single amino acid substitutions alone can generate broad functional diversity, highlighting the evolutionary adaptability of the RBP. Second, T7 can be optimized and activity can be increased, even on hosts that T7 is already considered to grow well on. Third, enrichment patterns on each host follow broad trends but have nuance at each position.

## Comparison across hosts reveals regions of functional importance

Next, we sought to elucidate features of each residue unique to each host or common across all hosts. There were over 30 residues with contrasting substitution patterns between different hosts, revealing fascinating features of receptor recognition for T7 (*Figure 2—source data 2*). Here, we focus on five of these residues, N501, R542, G479, D540, and D520, which showed starkly contrasting patterns of selection (*Figure 3A*). N501 and R542 are located on exterior loops oriented away from the phage and toward the receptor (*Figure 3C*). In fact, R542 forms a literal 'hook' to interact with the receptor (*Garcia-Doval and van Raaij, 2012*). On 10G and BL21, only positively charged residues (R, K, and H) were tolerated at residues 501 and 542, while in contrast many more substitutions were tolerated at both residues on BW25113. One such substitution, R542Q, is the best performing variant on BW25113 ($F_N$ = 3.31) but is conspicuously depleted on 10G and BL21, suggesting that even subtle molecular disparities can lead to large biases in activity. The substitution profiles of G479 and D540 are loosely the inverse of N501 and R542 as many substitutions are tolerated on BL21 and 10G, but very few are tolerated on BW25113 (*Figure 3A*). We hypothesize that D540 is critical for host recognition on BW25113. Since D540, a receptor-facing position on an exterior loop, is only 6 Å from G479, it is likely that any substitution at G479 may sterically hinder D540, resulting in the noted depletion of G479 substitutions on BW25113. This hypothesis is further supported by enrichment of adjacent S541D on BW25113 ($F_N$ = 2.82, the third highest scoring substitution), while this substitution is depleted on 10G and BL21 (*Figure 2—source data 2*). D520 displays a third variation in substitution patterns where substitutions are generally tolerated on 10G and BW25113, but not tolerated on BL21 (*Figure 3A*). This loop is also oriented downward toward the receptor, and we hypothesize that D520 or the local region around this exterior loop is more important for receptor recognition in BL21 than it is for the other two hosts, mirroring the result for D540 for BW25113. Another stark contrast can be drawn at adjacent S519, where no substitutions are tolerated in BL21 or 10G but several substitutions are enriched on BW25113, indicating that substitutions can improve receptor binding on one host while reducing function on another host. Overall, these host-specific substitution patterns reveal a nuanced relationship between the tip domain composition and receptor preferences.

We quantitatively characterized the role of every residue by integrating selection data across all hosts to reveal a functional map of the tip domain at granular resolution (*Figure 3B, C*, *Figure 3—source data 1*). We classified every residue as 'intolerant', 'tolerant', or 'functional' based on aggregated $F_N$ scores of all substitutions across all three hosts at every residue. Our method of classifying functional regions was robust to adjusting the $F_N$ threshold used to identify functional variants (*Figure 3—figure supplement 2*). Residues where the majority of substitutions were depleted were considered intolerant to substitution, while residues where at least a third of substitutions were depleted in one host and tolerated or enriched in another host were considered functional; the remaining positions were considered tolerant (see Materials and methods). The hydrophobic core comprising W474, I495, W496, I497, Y515, W523, L524, F526, I528, F535, and I548 is essential for stability and therefore is highly intolerant to substitutions (*Figure 3B*). Other intolerant positions include an elaborate network of salt bridge interactions involving D489, R491, R493, R508, and D512 in the interior loops, which likely constrain the orientation of the tip domain relative to the shaft (*Figure 3C*). Glycines generally provide conformational flexibility between secondary structure elements and normally tend to be mutable. Interestingly, several glycines (G476, G510, G522 ,and G532) are highly intolerant to substitutions. These glycines may be essential to minimize steric obstruction to adjacent larger residues, similar to G479 and D540 on BW25113 (*Figure 3C*). For example, G510 and G532 may facilitate formation of salt bridges in the interior loop, while G476 and G522 may facilitate a required receptor interaction in exterior loops for all three hosts.

It has been previously assumed that exterior loops are the primary functional region of the tip domain (*Garcia-Doval and van Raaij, 2012*; *Yehl et al., 2019*). We found that functional positions did typically point outward and are densely concentrated along exterior loops BC, DE, FG, and HI, as well as adjacent β-sheet residues. This is consistent with two specificity-switching substitutions found in a previous study, D520Q and V544A, which are both located in exterior loops (*Heineman et al., 2008*). However, several residues in exterior loops, such as G476 and S543, were notably intolerant, indicating that these residues may be poor targets for engineering or future combinatorial studies. Functional positions were also found in regions other than exterior loops, such as

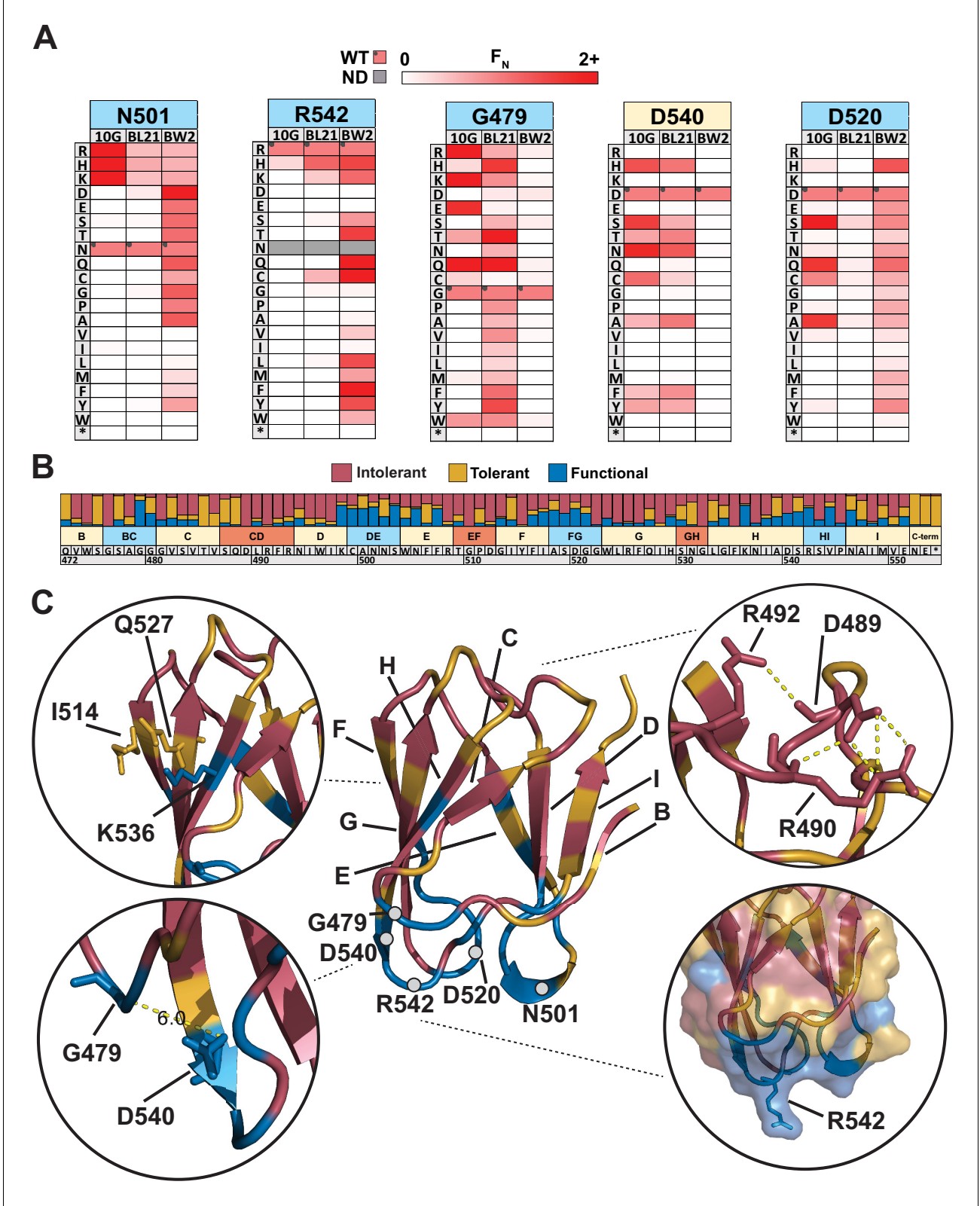

**Figure 3.** Comparison across hosts reveals regions of functional importance. (A) Host-specific differences in substitution patterns at five positions in the tip domain recapitulated from *Figure 2*. (B) Role of each position determined by aggregating scores of all substitutions in all hosts at that position. Substitutions are classified as intolerant ($F_N < 0.1$ in all hosts), tolerant ($F_N \geq 0.1$ in all hosts), or functional ($F_N < 0.1$ in one host, $F_N \geq 0.1$ in another host) and bar plots are shown as proportion of classified variants at that position. (C) Crystal structure of the tip domain (center) with each residue

*Figure 3 continued on next page*

*Figure 3 continued*

colored as intolerant, tolerant, or functional based on the dominant effect at that position, β-sheets and residues listed in (**A**) are labeled. Key interactions defining function and orientation are highlighted in peripheral panels.

The online version of this article includes the following source data and figure supplement(s) for figure 3:

**Source data 1.** Functional comparison for each variant on susceptible hosts.
**Figure supplement 1.** Comparing $F_D$ to computationally predicted stability of variant ΔΔG.
**Figure supplement 1—source data 1.** ΔΔG and $F_D$ conversion for all variants.
**Figure supplement 2.** Classification of tolerant, intolerant, and functional regions based on different cutoff conditions.
**Figure supplement 3.** Truth table comparing functional results to predicted stability.

I514, Q527, and K536, which are β-sheet residues located along one side of the tip domain (*Figure 3C*). This suggests the phage can use the 'side' of the tip domain to engage the receptor, increasing the apparent functional area of the tip domain and highlighting several new regions as valuable engineering targets.

We also determined if the functionally important regions could be predicted computationally as the ability to predict functionally important regions without DMS could rapidly accelerate engineering efforts. We used Rosetta, a state-of-the-art protein modeling software, to calculate the change in Gibbs free energy (ΔΔG) for each of the 1660 mutations and compared this distribution to our DMS results (*Figure 3—figure supplement 1*, also see Materials and methods) and generated a truth table to summarize results compared to our functional data (*Figure 3—figure supplement 3*). Predicted thermodynamic changes in stability mapped very well with over 93% of tolerated or functional positions having a substitution that was predicted to be stabilizing. The remaining 7% of tolerant or enriched substitutions were predicted to be destabilizing, and we hypothesize that this may indicate these substitutions result in improved dynamic or induced fit positioning of the tip domain for productive infection. Incorporating stability estimations could further improve the engineering power of the assay. For example, substitutions predicted to be stable but that are intolerant in the DMS assay may indicate that the substituted residue is necessary for all three hosts.

Overall, these results paint a complex enrichment profile for each host with some broad trends but subtle host-specific effects. These results suggest that exterior loops and some outward-facing positions in β-sheets act as a reservoir of function-switching and function-enhancing mutations, likely promoting host-specific and orientation-dependent interactions between phage and bacterial receptors. Functional positions identified by this comparison are ideal engineering targets to customize host range and activity.

## Discovery of gain-of-function variants against resistant hosts

The tail fiber is considered a reservoir of gain-of-function variants due to its principal role in determining fitness of a phage through host adsorption (*Holtzman et al., 2020*; *Yehl et al., 2019*). We hypothesized that novel gain-of-function variants against a resistant host could be discovered by subjecting our tail fiber variant library to selection on a resistant host. To identify a resistant host, we focused on host genes *rfaG* and *rfaD* involved in the biosynthesis of surface LPS, which is a known receptor for T7 in *E. coli* (*González-García et al., 2015*; *Molineux, 2001*; *Qimron et al., 2006*). Gene *rfaG* (synonyms *WaaG* or *pcsA*) transfers glucose to the outer core of LPS and deletion strains lack the outer core of LPS (*Pagnout et al., 2019*), while *rfaD* (synonyms *gmhD* or *WaaD*) encodes a critical epimerase required for building the inner core of LPS (*Valvano et al., 2002*; *Figure 4A*). Deletion of either gene reduces the ability of T7 to infect *E. coli* by several orders of magnitude (*Figure 4F*). We challenged the library of T7 variants against *E. coli* deletion strains BW25113Δ*rfaG* and BW25113Δ*rfaD* through pooled selection and deep sequencing as before (*Figure 2*) and determined an $F_N$ score for each substitution on both strains (*Figure 4B, C*). Independent replicates showed good correlation for BW25113Δ*rfaG* (R = 0.99, 0.93, 0.93) but only adequate correlation for BW25113Δ*rfaD* (R = 0.51, 0.68, 0.39) (*Figure 4—figure supplement 1*). Although the scale of $F_N$ was inconsistent across replicates on BW25113Δ*rfaD*, the same substitutions were largely enriched in all three replicates, suggesting reproducibility of results (*Figure 4—figure supplement 3*). Inconsistencies in $F_N$ scores may arise due to severe loss of diversity causing stochastic differences in enrichment to become magnified across independent experiments and the four infection cycles used for

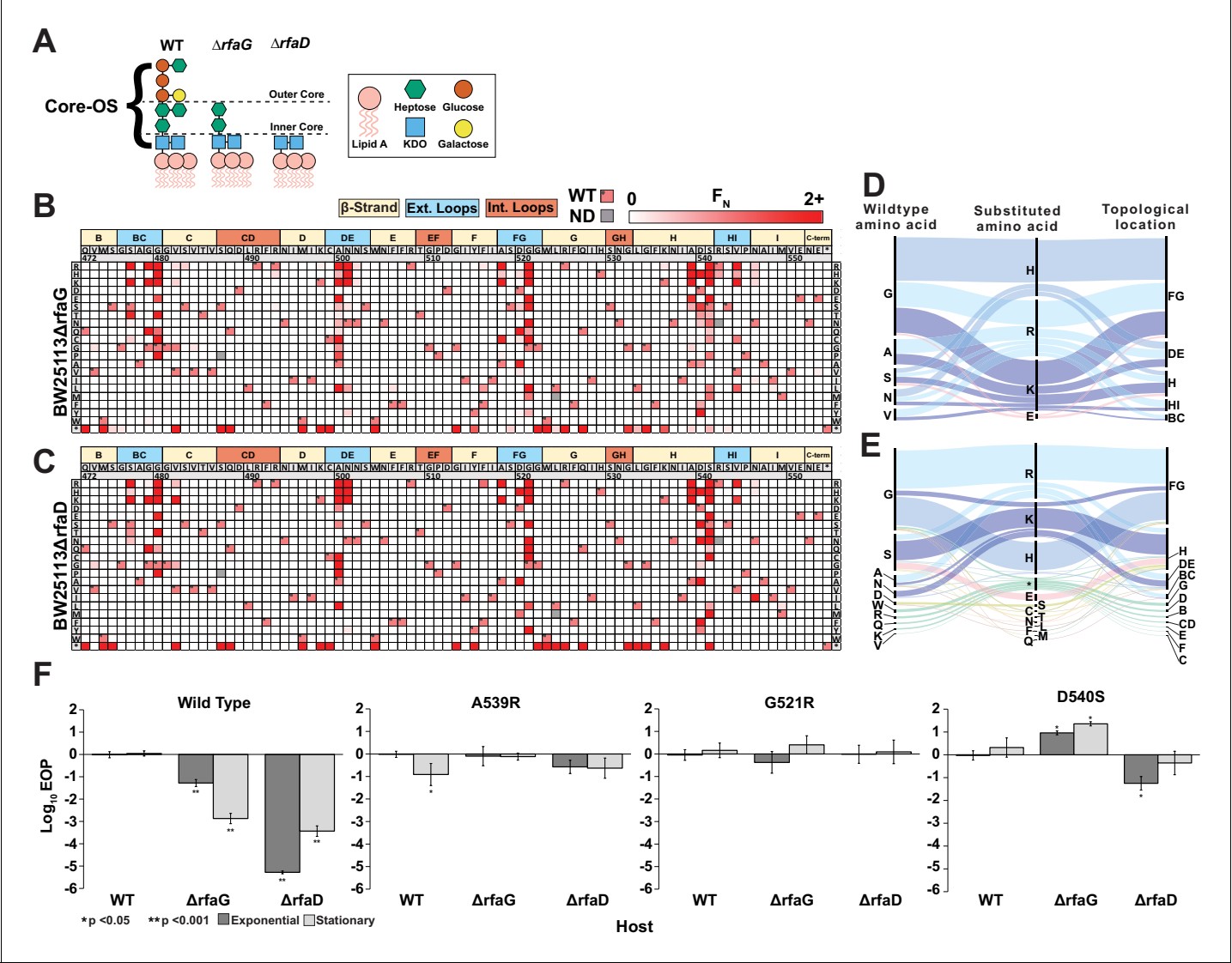

**Figure 4.** Discovery of gain-of-function variants against resistant hosts. (A) Schematic view of the LPS on wildtype BW25113, BW25113Δ*rfaG* and BW25113Δ*rfaD*. (B-C) Heat maps showing normalized functional scores ($F_N$) of all substitutions (red gradient) and wildtype amino acid ($F_N$=1 and black dot upper left) at every position for BW25113Δ*rfaG* (B) and BW25113Δ*rfaD* (C). (D-E) Among highly enriched variants ($F_N \geq 10$), targeted amino acids (left), their substitutions (middle) and topological location on the structure (right) on BW25113Δ*rfaG* (D) and BW25113Δ*rfaD* (E), with each alluvial colored based on the substituted amino acid and scaled by $F_N$. (F) EOP (mean ± SD, biological triplicates) for wildtype phage and select variants on BW25113 (Wild Type), BW25113Δ*rfaG* and BW25113Δ*rfaD* in exponential (dark gray) and stationary phases (light gray) using BW25113 as a reference host.

The online version of this article includes the following source data and figure supplement(s) for figure 4:

**Source data 1.** Deep sequencing summary for phage variant library after selection on different hosts.

**Source data 2.** Variant-specific $F_N$ for phage variants after selection on *E. coli* BW25113Δ*rfaG* and BW25113Δ*rfaD* and physicochemical statistics.

**Figure supplement 1.** Correlation between biological replicates after selection of phage variant library on *E. coli* BW25113Δ*rfaG* and BW25113Δ*rfaD*.

**Figure supplement 2.** Distribution, enrichment profile, and physicochemical properties of variants after selection on *E. coli* BW25113Δ*rfaG* and BW25113Δ*rfaD*.

**Figure supplement 3.** Ranking $F_N$ of the 10 most enriched variants in each biological replicate for (A) *E. coli* BW25113ΔrfaG and (B) BW25113Δ*rfaD*.

**Figure supplement 4.** Correlation between biological replicates for selection of phage variant library after one infection cycle on (A) *E. coli* BW25113Δ*rfaG* and (B) BW25113Δ*rfaD*.

**Figure supplement 5.** Location of truncations in the tip domain enriched after selection.

selection. Separately we examined correlation after selection using only a single infection cycle, which produced more highly correlated results for BW25113Δ*rfaD* (R = 0.89, 0.90, 0.89) (*Figure 4—figure supplement 4*), indicating that fewer infection cycles may be ideal for future work with highly resistant hosts.

We engineered several gain-of-function T7 variants that could infect both deletion strains with activity comparable to wildtype T7 infecting susceptible BW25113 (*Figure 4F*). Low-sequence diversity and high enrichment scores of T7 variants indicate a strong selection bottleneck, which is consistent with diminished activity of wildtype T7 on the deletion strains. This is reflected in the significantly lower functional score of wildtype T7 on BW25113Δ*rfaG* and BW25113Δ*rfaD* (F = 0.09 ± 0.3 and F = 0.03 ± 0.2, respectively) in comparison to BW25113 (F = 2.26 ± 0.1, p < 0.001) (*Figure 2—source data 1*). The number of enriched variants outperforming wildtype T7 ($F_N \geq 2$) on the deletion strains (BW25113Δ*rfaG*: 55 variants, 3.3% and BW25113Δ*rfaD*: 68 variants, 4.1%) was over three times higher than BW25113 (16 variants, 1%) but comparable to 10G (48 variants, 2.9%) (*Figure 4—figure supplement 2A, B*). However, the enrichment scores of top performing variants such as G521H and G521R on BW25113Δ*rfaG* and S541K and N501H on BW25113Δ*rfaD* were over 100 times greater than wildtype T7, suggesting strong gain of function on the deletion strains (*Figure 4—figure supplement 2C*). Of the 78 variants with $F_N \geq 2$ on either deletion strain, 45 variants had $F_N \geq 2$ on both strains, indicating that variants that performed well on one strain typically performed well on the other strain. This implies that the enriched variants may have broad affinity for truncated LPS but cannot discriminate based on the length of the LPS. Nonetheless, hydrophilic substitutions were more strongly enriched on *BW25113Δrfa*G (p < 0.001, r > 0.11), but not as significantly on *BW25113Δrfa*D (p < 0.033, r < 0.10), suggesting subtle differences in surface chemical properties of deletion strains leading to host-specific enrichments (*Figure 4—figure supplement 2D–F*). Indeed, there were several variants with contrasting F scores on both strains such as S541T (*BW25113Δrfa*D $F_N$ = 44.8, *BW25113Δrfa*G $F_N$ = 0.6) and G521E (*BW25113Δrfa*D $F_N$ = 0, *BW25113Δrfa*G $F_N$ = 17.4), suggesting potential host preference. Most substitutions were concentrated in the exterior loops BG, FG, HI, and β-strand H, all pointing downward toward the bacterial surface, reinforcing the functional importance of these regions of the tip domain (*Figure 4D, E*). Notably, the most enriched variants had large positively charged substitutions (K, R, and H) akin to the enrichment pattern on 10G, suggesting that the bacterial surface of these truncated mutants likely resembles that of 10G. Our results are consistent with a recent continuous evolution study, which identified G480E and G521R as possible gain-of-function variants on a strain similar to *BW25113Δrfa*D and G479R and G521S as possible gain-of-function variants on *BW25113Δrfa*G (*Holtzman et al., 2020*), although these variants only represent a small fraction of the gain-of-function variants discovered in our study.

We validated the results of the pooled selection experiment by clonally testing the ability of phage variants with high $F_N$ (A539R, G521H, and D540S) to plaque on the deletion strains based on a standard efficiency of plating (EOP) assay. Indeed, EOP results showed significant gain of function in these variants on the deletion strains (*Figure 4F*). D540S was particularly noteworthy as it performed better on the deletion strain *BW25113Δrfa*G over wildtype BW25113 by 1–2 orders of magnitude. Based on these results, we conclude that D540 is critical for infecting wildtype BW25113 (*Figure 3*) likely by interacting with the outer core of LPS. When the outer core of the LPS is missing (*BW25113Δrfa*G), a substitution at this position becomes necessary for adsorption either to a different LPS moiety or to an alternative receptor.

We introduced stop codon at every position to systematically evaluate the function of tip domains truncated to different lengths. Many truncated variants performed well, especially on BW25113Δ*rfaG*, which included some with $F_N \geq 10$ (*Figure 4—source data 2*). Truncated variants that performed well are distributed throughout the tip domain and are not localized to any one region (*Figure 4—figure supplement 5*). We clonally tested variant R525*, the best performing truncated library member (BW25113Δ*rfaG* $F_N$ = 9.55, BW25113Δ*rfaD* $F_N$ = 75.7), and found that this mutant showed no ability to plaque on any host unless provided the tail fiber in trans. These truncated phages, detectable here only using deep sequencing, may demonstrate how obligate lytic phages could become less active in a bacterial population, slowly replicating alongside their bacterial hosts, requiring only a single mutation to become fully active again. In fact, acceptor phages altogether lacking a tail fiber were present at extremely low abundance (*Figure 4—source data 1*). These phages are not artifacts from library creation as some ability to replicate is required to

produce detectable concentrations of each phage. We concluded that these are viable phage variants albeit with a much slower infection cycle, resulting in their inability to form visible plaques.

## Targeting pathogenic *E. coli* causing UTIs using T7 variants

Phage therapy is emerging as a promising solution to the antibiotic resistance crisis. Recent clinical success stories against multidrug-resistant *Acinetobacter* and *Mycobacterium* showcase the enormous potential of phage therapy (*Dedrick et al., 2019*; *Schooley et al., 2017*). Despite notable exceptions, in general development of effective phage-based therapeutics is hindered by onset of bacterial resistance, resulting in low phage susceptibility. Although initial application of phages in a laboratory setting may reduce bacterial levels, the residual bacterial load remains high, causing bacteria to quickly recover after phage application (*Fister et al., 2016*; *Huss and Raman, 2020*; *Silva et al., 2014*). A high ratio of phage to bacteria (multiplicity of infection [MOI]) may productively kill bacteria in a laboratory setting by overwhelming a host with many phages (*Abedon, 2011*). However, ensuring an overwhelming amount of phages in a clinical setting is not always feasible (*Principi et al., 2019*). Engineering highly active phages that overcome bacterial insensitivity and can therefore productively eliminate bacterial populations at low MOI in a laboratory setting would greatly enhance phage-based therapeutics. We hypothesized that engineered tip domain variants may abate bacterial insensitivity and be active even at low MOI by better adsorbing to the native receptor or recognizing a new bacterial receptor altogether.

To test this hypothesis, we chose pathogenic *E. coli* strain isolate 473 isolated from a patient with a UTI (*Arthur et al., 1990*). Although T7 can infect this UTI strain, insensitivity arises rapidly, a phenomenon all too common with the use of natural phages. EOP assays for wildtype T7 showed insensitive plaque morphology consisting of small, slow-growing plaques. No visible lysis was detected after overnight incubation when wildtype T7 was applied in liquid culture (MOI = 1), indicating onset of insensitivity. However, the variant library applied on the UTI strain cleared the culture (MOI = 1) after overnight incubation, suggesting that T7 variants are capable of lysing and attenuating insensitivity existing in the pool. We clonally characterized three variants (N501H, D520A, and G521R) isolated from plaques. All three variants vastly outperformed wildtype T7 in terms of onset of insensitivity. Insensitivity emerged approximately 11–13 hr after initial lysis for the three variants, whereas it took merely 1–2 hr after initial lysis for wildtype (*Figure 5A*). In particular, the N501H variant lysed cells faster and produced a lower bacterial load post lysis, suggesting far greater activity compared to wildtype T7. Next, we compared the effect of phage MOI (MOI = $10^2$-$10^{-5}$) on the lysing activity of N501H and wildtype T7 (*Figure 5B*). At all MOIs, wildtype phages lysed UTI473 significantly more slowly compared to N501H phages (all p < 0.05). At lower MOI, time to lysis of N501H was half that of wildtype T7, though they were more comparable at higher MOI.

A striking contrast between N501H and wildtype T7 is evident in reduced bacterial insensitivity at progressively lower MOI (*Figure 5C*). Between an MOI of 100 to 1, application of both N501H and wildtype phage resulted in similar bacterial insensitivity. However, between an MOI of $10^{-1}$ and $10^{-5}$, application of N501H phage reduced insensitivity over a 10 hr window, while application of wildtype phages resulted in rapid onset of insensitive bacteria. We postulate that at high MOI wildtype T7 simply overwhelms the host before insensitivity arises, while at lower MOI insensitivity can emerge and only variants adapted to the host can effectively kill the host. These results indicate that ORACLE can generate phage variants superior to wildtype phage that could then become starting points for further engineering therapeutic phages. Further experiments will be required to assess the in vivo efficacy of the T7 variants.

## Host range constriction emerges from global comparison across variants

Most phages are specialists that selectively target a narrow range of hosts but are unable to productively infect other closely related hosts (*Hyman and Abedon, 2010*). We wanted to assess differences in the host range of individual variants on 10G, BL21, and BW25113 and identify variants with constricted host ranges. Ideally, host specificities can be determined by subjecting a co-culture of all three hosts to the phage library. However, deconvolving specificities of thousands of variants from a pooled co-culture experiment can be technically challenging. Instead, we sought to estimate specificities by comparing $F_N$ of a phage variant on all three hosts. Although $F_N$ compares activity of

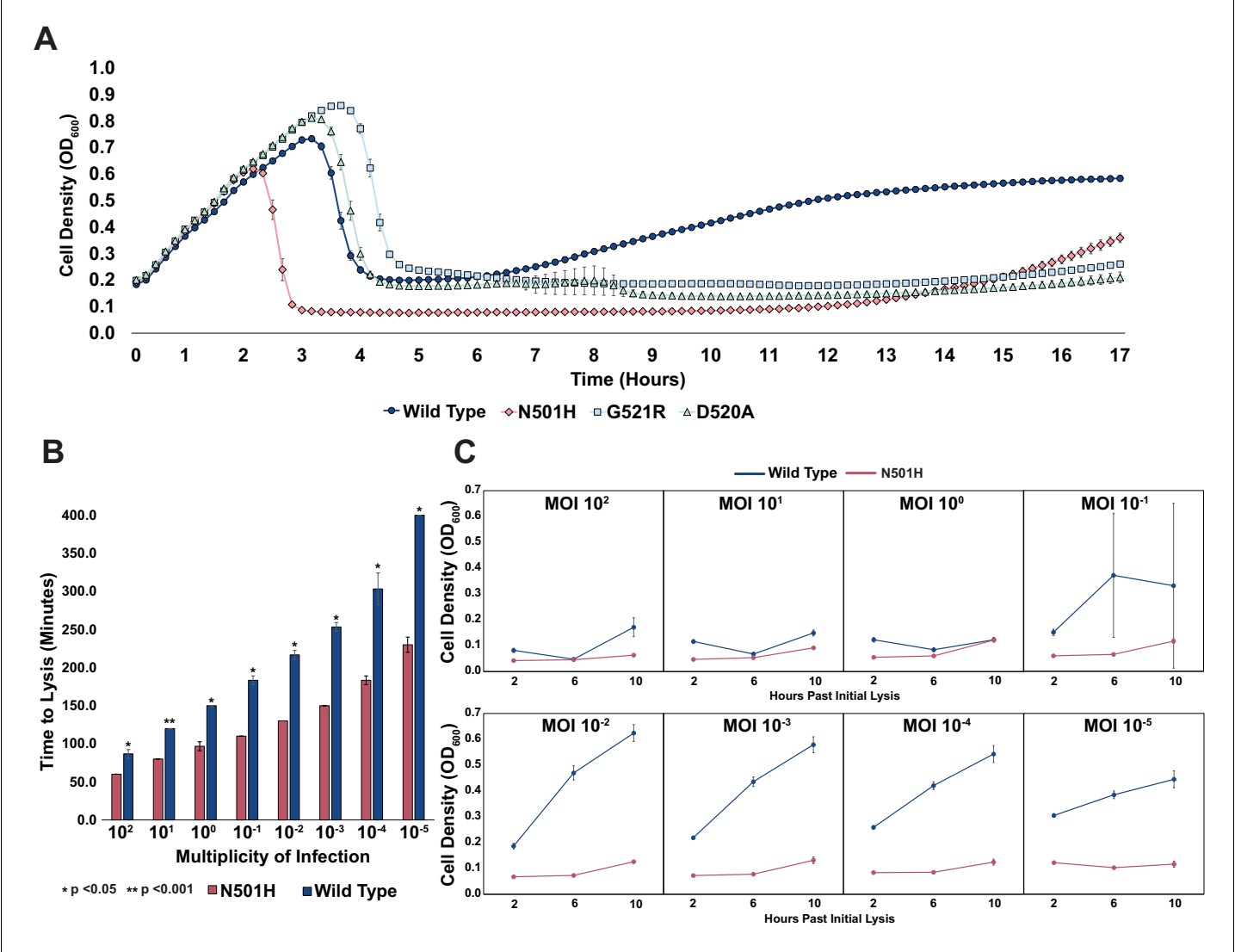

**Figure 5.** Targeting pathogenic *E. coli* causing UTIs using T7 variants. (**A**) Growth time course of UTI473 strain subject to wildtype T7 and select variants. Phages were applied after an hour at an MOI of ~$10^{-2}$. (**B**) Estimated time to lysis of UTI473 strain incubated with wildtype T7 and N501H variant over a range of MOIs, derived from time course experiments. (**C**) Cell density ($OD_{600}$) of UTI473 strain when incubated with wildtype T7 and N501H variant at select timepoints after initial lysis. All data represented as mean ± SD of biological triplicate.

variants within a host, it could nonetheless be a useful proxy for estimating specificities across hosts. For instance, a phage variant with high $F_N$ on BL21 but completely depleted on BW25113 is more likely to specifically lyse BL21 than BW25113 in a co-culture experiment. Based on this rationale, we considered different metrics of comparison of $F_N$ and settled on difference in $F_N$ of a variant with reduced weight for enrichment (or $F_D$, see Materials and methods) between any two hosts as an approximate measure of host preference. This metric is not an absolute measure of host specificity, but one devised to reveal broad trends in specificity to prioritize variants for downstream validation.

To assess if variants preferred one host over another, we computed $F_D$ for all three pairwise combinations and plotted functional substitutions as points on or above/below a 'neutral' line (*Figure 6A–C*). Variants above the line favor lysis of the noted host, and vice versa for variants below the line. To check if this $F_D$-based approach is suitable for assessing host specificity, we compared our results with previously published data. Two substitutions, D520Q and V544A, that were reported to have a preference for BW25113 and BL21, respectively, in head-to-head comparisons (*Heineman et al., 2008*) were placed correctly in our plots, confirming the validity of our $F_D$-based

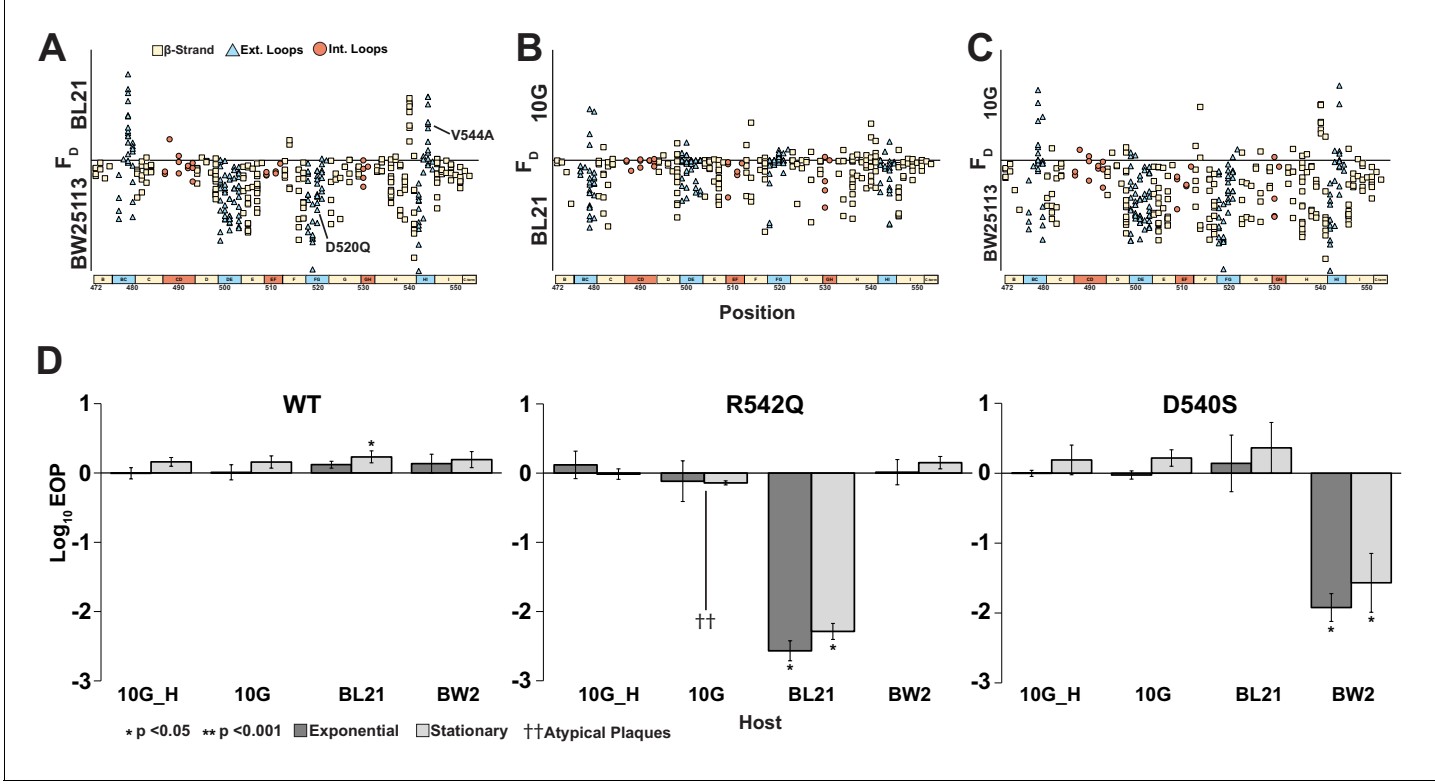

**Figure 6.** Host range constriction emerges from global comparison across variants. (A-C) Pairwise comparison of differences in functional scores of variants between hosts (see Methods). Variants above the line favor lysis of host noted above the line, and vice versa for variants below the line. (D) EOP (mean ± SD, biological triplicates) for wildtype T7 and select variants on BW25113, 10G and BL21 in exponential (dark gray) and stationary phases (light gray) using exponential 10G with gp17 tail fiber helper plasmid (10G_H) as a reference host. R542Q plaques are atypically small until EOP ~10-2. The online version of this article includes the following source data for figure 6:

**Source data 1.** ΔΔG and $F_D$ conversion for all variants.

classification scheme. We identified 118 out of 1660 variants as good candidates for constricting host range ($|F_D| \geq 1$, see ***Figure 6—source data 1***). Of the 118 variants, 53 variants favor BW25113 over BL21 and 98 variants favor BW25113 over 10G in pairwise comparisons (***Figure 6A, C***). Between BL21 and 10G, there are 15 variants that favor BL21 but none that favor 10G (***Figure 6B***).

Certain key positions, including G479, D540, R542, and D520, which we previously identified as functionally important (***Figure 3A***), are the molecular drivers of specificity between hosts (***Figure 6A–C***). Taken together, our data suggests that it would be easier to find a variant capable of specifically lysing BW25113, less so for BL21, and most challenging for 10G.

To validate our analysis, we clonally tested variant R542Q, which had a greater preference for BW25113 than BL21 or 10G (BW25113 $F_D$ = 2.0, BL21 $F_D$ = 0, 10G $F_D$ = 0), and variant D540S, which had a greater preference for BL21 and 10G than BW25113 (10G $F_D$ = 1.03, BL21 $F_D$ = 0.62, BW25113 $F_D$ = 0.03). Indeed, R542Q showed a significant approximately hundredfold decrease in the ability to plaque on BL21 compared to BW25113 while 10G plaques were atypically small, indicating a severe growth defect (***Figure 6D***). In contrast, D540S showed a significant approximately hundredfold decrease in the ability to plaque on BW25113 compared to BL21 and 10G (***Figure 6D***), confirming the host constriction properties of these variants. In summary, pairwise comparison is a powerful tool to map substitutions that constrict host range and can be leveraged to tailor engineered phages for targeted hosts.

## Discussion

In this study, we used ORACLE to create a large, unbiased library of T7 phage variants to comprehensively characterize the mutational landscape of the tip domain of the tail fiber. Our study identified hundreds of novel function-enhancing substitutions that had not been previously characterized. We mapped regions of function-enhancing substitutions on to the crystal structure to rationalize how sequence and structure influence activity and host range. Several important insights emerged from these results. Cross-comparison between different hosts and selection on resistant hosts allowed us to map key substitutions, leading to host discrimination and gain of function. Single amino acid substitutions are sufficient to enhance activity and host range, including some that confer dramatic increases in activity or specificity. The functional landscape on each host is unique, reflecting both different molecular preferences of adsorption and the fitness of wildtype T7 on these hosts. For instance, hydrophilic substitutions were enriched in 10G while hydrophobic substitutions were enriched in BW25113. Notably, substitutions on 10G (an *E. coli* K-12 derivative lacking LPS components) mirrored substitutions that recovered function on BW25113 mutants with truncated LPS, which shows convergence of selection. Function-enhancing substitutions were densely concentrated in the exterior loops, indicating an orientational preference for receptor recognition. However, they were also found on other surface residues, albeit less frequently, suggesting alternative binding modes of the tip domain for host recognition, and several intolerant residues were located in exterior loops. Taken together, these results highlight the extraordinary functional potential of the tip domain and rationalize the pervasive use of this structural fold in nature for molecular recognition. Comparison of these functional profiles precisely reveals the regions that are ideal engineering targets for customizing host range and activity and identifies intolerant residues that should be avoided when engineering synthetic phages.

These results also highlight the power of deep sequencing to detect and resolve small functional effects over traditional low-throughput plaque assays. This is best illustrated in the case of truncated variants visible only to deep sequencing, but incapable of plaque formation without a helper plasmid. The truncated variants are likely not experimental artifacts as some ability to replicate is required to survive multiple rounds of selection on the host. Truncation of the tip domain may misorient the phage relative to the receptor, likely resulting in slower growth and deficiency in plaquing, while still capable of replicating. Since plaque formation is a complex process, inability to plaque may not imply a functionally incompetent phage.

ORACLE is designed as a foundational technology to elucidate sequence–function relationships in phage genes. On T7, ORACLE can be used to investigate the function of several important genes, including the remainder of the tail fiber and tail structure, capsid components, or lysins and holins. Together, these will provide a comprehensive view of the molecular determinants of the structure, function, and evolution of a phage. Once the phage variants are created, scaling up ORACLE to investigate potentially tens of hosts merely scales up sequencing volume, not experimental complexity. Such a large-scale study will lead to a detailed molecular understanding and adaptability of phage bacterial interactions. Any phage with a sequenced genome and a transformable host capable of maintaining a plasmid library should be amenable to ORACLE because the phage variants are created during the natural infection cycle. This approach can be leveraged to tune activity for known phages with high activity, such as T7, or to identify engineering targets that dramatically increase activity for newly isolated natural phages.

The confluence of genome engineering, high-throughput DNA synthesis, and sequencing enabled by ORACLE together with viral metagenomics could transform phage biology. Phages constitute unparalleled biological variation found in nature and are aptly called the 'dark matter' of the biosphere. Their sequence diversity and richness are coming to light in the growing volume of viral metagenome databases. However, what functions these sequences encode remains largely unknown. For instance, fecal viromes estimate $10^8$–$10^9$ virus-like particles per gram of feces, but less than a quarter of sequence reads align to existing databases (*Reyes et al., 2012*). While this knowledge gap is daunting, it also presents an opportunity to mine metagenomic sequences to characterize their function and engineer programmable phages. By enabling sequence programmability, we envision ORACLE as a powerful tool to discover new phage 'parts' from metagenomic sequences.

# Materials and methods

**Key resources table**

| Reagent type (species) or resource | Designation | Source or reference | Identifiers | Additional information |
|---|---|---|---|---|
| Strain, strain background (*Escherichia coli*) | *E. coli* 10G | Lucigen | Lucigen:60107-1 | |
| Strain, strain background (*Escherichia coli*) | *E. coli* BL21 | ATCC | ATCC:BAA-1025 | |
| Strain, strain background (*Escherichia coli*) | *E. coli* 10-beta | NEB | NEB:C3020 | |
| Strain, strain background (*Escherichia coli*) | *E. coli* BW25113 | *Baba et al., 2006* | BW25113 | |
| Strain, strain background (*Escherichia coli*) | *E. coli* BW25113Δ*rfaG* | *Baba et al., 2006* | BW25113Δ*rfaG* | |
| Strain, strain background (*Escherichia coli*) | *E. coli* BW25113 Δ*rfaD* | *Baba et al., 2006* | BW25113 Δ*rfaD* | |
| Strain, strain background (*Escherichia coli*) | *E. coli* UTI473 | *Arthur et al., 1990* | UTI473 | |
| Strain, strain background (T7 bacteriophage) | T7 bacteriophage | ATCC | ATCC:BAA-1025-B2 | |
| Strain, strain background (T7 bacteriophage) | T7 bacteriophage variants | This paper | Available on request | DMS variants, available from the Raman lab. |
| Commercial assay or kit | KAPA HiFi PCR Kit | Roche | Roche:KK2101 | |
| Commercial assay or kit | KAPA2G Robust PCR Kit with dNTPS | Roche | Roche:KK5005 | |
| Commercial assay or kit | Golden Gate Assembly Kit (BsaI-HFv2) | NEB | NEB:E1601L | |
| Recombinant DNA reagent | pHT7Helper1 (plasmid) | This paper | | Helper with T7 gp17. See Materials and methods for full details. |
| Recombinant DNA reagent | pHRec1 and derivatives (plasmids) | This paper | | Recombination plasmid. See Materials and methods for details. |
| Recombinant DNA reagent | pHCas9 and derivatives (plasmids) | This paper | | Plasmid with Cas9 targeting acceptor phage. See Materials and methods for full details. |
| Software, algorithm | R scripts for DMS analysis | This paper | N/A | Available here https://github.com/raman-lab/oracle; *Huss, 2021*; copy archived at swh:1:rev:657e8eef12e4ee886f5d188b745ff0b38f94f479 |
| Software, algorithm | R scripts for physicochemical comparisons | This paper | N/A | Available here https://github.com/raman-lab/oracle. |
| Software, algorithm | R scripts for Rosetta ΔΔG calculations | This paper | N/A | Available here https://github.com/raman-lab/oracle. |

## Microbes and culture conditions

T7 bacteriophage was obtained from ATCC (ATCC BAA-1025-B2). *Saccharomyces cerevisiae* BY4741, *E. coli* BL21 is a lab stock, *E. coli* 10G is a highly competent DH10B derivative (*Durfee et al., 2008*) originally obtained from Lucigen (60107-1). *E. coli* 10-beta was purchased from NEB (C3020). *E. coli* BW25113, BW25113Δ*rfaD*, and BW25113Δ*rfaG* were obtained from Doug Weibel (University of Wisconsin, Madison) and are derived from the Keio collection (*Baba et al., 2006*). UTI473 was obtained from Rod Welch (University of Wisconsin, Madison) and originates from a UTI collection (*Arthur et al., 1990*).

All bacterial hosts are grown in and plated on Lb media (1% tryptone, 0.5% yeast extract, 1% NaCl in dH$_2$O, plates additionally contain 1.5% agar, while top agar contains only 0.5% agar), and Lb media was used for all experimentation. Kanamycin (50 µg/ml final concentration, marker for pHT7Helper1) and spectinomycin (115 µg/ml final concentration, marker for pHRec1, pHRec1-Lib, and pHCas9 and derivatives) were added as needed. All incubations of bacterial cultures were performed at 37°C, with liquid cultures shaking at 200–250 rpm unless otherwise specified. Bacterial hosts were streaked on appropriate Lb plates and stored at 4°C.

*S. cerevisiae* BY4741 was grown on YPD (2% peptone, 1% yeast extract, 2% glucose in dH$_2$O, plates additionally contain 2.4% agar), after yeast artificial chromosomes (YAC) transformation. *S. cerevisiae* BY4741 was grown on SD-Leu (0.17% yeast nitrogen base, 0.5% ammonium sulfate, 0.162% amino acids – leucine [Sigma Y1376], 2% glucose in dH$_2$O, plates additionally contain 2% agar). All incubations of *S. cerevisiae* were performed at 30°C, with liquid cultures shaking at 200–250 rpm. *S. cerevisiae* BY4741 was streaked on YPD or SD-Leu plates as appropriate and stored at 4°C.

T7 bacteriophage was propagated using *E. coli* BL21 after initial receipt from ATCC and then as described on various hosts in methods. All phage experiments were performed using Lb and culture conditions as described for bacterial hosts. Phages were stored in Lb at 4°C.

For long-term storage, all microbes were stored as liquid samples at −80°C in 10% glycerol, 90% relevant media.

SOC (2% tryptone, 0.5% yeast extract, 0.2% 5 M NaCl, 0.25% 1 M KCl, 1% 1 M MgCl$_2$, 1% 1 M MgSO$_4$, 2% 1 M glucose in dH$_2$O) was used to recover host and phages after transformation.

## General cloning methods

PCR was performed using KAPA HiFi (Roche KK2101) for all experiments with the exception of multiplex PCR for screening YACs, which was performed using KAPA2G Robust PCR kits (Roche KK5005). Golden Gate Assembly was performed using New England Biosciences (NEB) Golden Gate Assembly Kit (BsaI-HFv2, E1601L). Restriction enzymes were purchased from NEB with the exception of DNAse I (Roche 4716728001). DNA purification was performed using EZNA Cycle Pure Kits (Omega Bio-tek D6492-01) using the centrifugation protocol. YAC extraction was performed using YeaStar Genomic DNA Extraction kits (Zymo Research D2002). Gibson assembly was performed according to the Gibson Assembly Protocol (NEB E5510), but Gibson Assembly Master Mix was made in lab (final concentration 100 mM Tris-HCl pH 7.5, 20 mM MgCl$_2$, 0.2 mM dATP, 0.2 mM dCTP, 0.2 mM dGTP, 10 mM dTT, 5% PEG-8000, 1 mM NAD$^+$, 4 U/ml T5 exonuclease, 4 U/µl Taq DNA ligase, 25 U/ml Phusion polymerase). All cloning was performed according to manufacturer's documentation except where noted in methods. If instructions were variable and/or specific conditions are relevant for reproducing results, those conditions are also noted in the relevant methods section.

PCR reactions use 1 µl of ~1 ng/µl plasmid or ~0.1 ng/µl DNA fragment as template for relevant reactions. PCR reactions using phage as template use 1 µl of undiluted or 1:10 diluted phage stock, genomic extraction was unnecessary. Phage template was initially treated at 65°C for 10 min (for YAC cloning), but we later simply extended the 95°C denaturation step to 5 min (for deep sequencing).

DpnI digest was performed on all PCR that used plasmid as template. Digestion was performed directly on PCR product immediately before purification by combining 1–2 µl DpnI (20–40 units), 5 µl 10× CutSmart Buffer, PCR product, and dH$_2$O to 50 µl, incubating at 37°C for 2 hr then heat inactivating at 80°C for 20 min.

DNAse treatment of phages was performed by adding 5 µl undiluted phages, 2 µl 10× DNAse I buffer, 1 µl of 2 U/µl DNAse I, dH$_2$O to 20 µl, then incubating for 20 min at 37°C, followed by heat inactivation at 75°C for 10 min. Then, 1 µl of this reaction was used as template for relevant PCR.

Electroporation of plasmids and YACs was performed using a Bio-rad MicroPulser (165-2100), Ec2 setting (2 mm cuvette, 2.5 kV, one pulse) using 25–50 µl competent cells and 1–2 µl DNA for transformation. Electroporated cells were immediately recovered with 950 µl SOC, then incubated at 37°C for 1–1.5 hr and plated or grown in relevant media.

*E. coli* 10G competent cells were made by adding 8 ml overnight 10G cells to 192 ml SOC (with antibiotics as necessary) and incubating at 21°C and 200 rpm until ~OD$_{600}$ of 0.4 as determined using an Agilent Cary 60 UV-Vis Spectrometer using manufacturer's documentation (actual incubation time varies based on antibiotic, typically overnight). Cells are centrifuged at 4°C, 800–1000 *g* for 20 min,

the supernatant is discarded, and cells are resuspended in 50 ml 10% glycerol. Centrifugation and washing are repeated three times, then cells are resuspended in a final volume of ~1 ml 10% glycerol and are aliquoted and stored at −80°C. Cells are competent for plasmid and YACs.

Site-directed mutagenesis (SDM) was performed, in brief, using complementary primers with the desired mutation in the middle of the primer, using 16× cycles of PCR, followed by DpnI digestion and electroporation into competent *E. coli* 10G. Splicing by overlap extension (SOE, also known as PCR overlap extension) was performed, in brief, using equimolar ratios of fragments, 16× cycles of PCR using extension based on the combined length of fragments, then a second PCR reaction using 1/100 or 1/1000 diluted product of the first reaction in a typical PCR reaction using 5′ and 3′ end primers for each fragment.

Detailed protocols for cloning are available on request. All primers used in experiments in this publication are listed in **supplementary file 1**.

## Plasmid cloning and descriptions

pHT7Helper1 contains a pBR backbone, kanamycin resistance cassette, mCherry, and the T7 tail fiber *gp17*. Both mCherry and *gp17* are under constitutive expression. *Gp17* was combined with promoter apFAB47 (**Kosuri et al., 2013**) using SOE and the plasmid assembled by Gibson assembly. There is a single-nucleotide deletion in the promoter that has no effect on plaque recovery for phages that require *gp17* to plaque. This plasmid is used during optimized recombination and accumulation in ORACLE to prevent library bias and depletion of variants that grow poorly on *E. coli* 10G. pHRec1 contains an SC101 backbone, Cre recombinase, a spectinomycin resistance cassette, and the T7 tail fiber *gp17* flanked by Cre lox66 sites with an m2 spacer, a 3′ pad region, and lox71 sites with a wt spacer (**Langer et al., 2002**; **Figure 1—figure supplement 1**). Cre recombinase is under constitutive expression. This plasmid was assembled with sequential PCR and Gibson assembly. During assembly, we used PCR overhangs and SDM to create two synonymous substitutions in *gp17* to remove two BsaI restriction sites, facilitating downstream Golden Gate Assembly. This plasmid was used in recombination assays as it allows for recombination of wildtype *gp17* and is used as template to generate the DMS variant library. The DMS variant library is referred to as pHRec1-Lib and is used during optimized recombination in ORACLE. Note that this assembly was not tolerated in higher copy number plasmids.

pHCas9 contains an SC101 backbone, a spectinomycin resistance cassette, and cas9 cassette capable of ready BsaI cloning of gRNA (**Jiang et al., 2013**). This plasmid is used directly as part of the negative control for the accumulation assay and has five derivatives, pHCas9-1 through -5, each with a different gRNA targeting the fixed region in the T7 acceptor phage. pHCas9 was created with Gibson assembly, while derivatives were assembled by phosphorylation and annealing gRNA oligos (100 µM forward and reverse oligo, 5 µl T4 Ligase Buffer, 1 µl T4 PNK, to 50 µl dH$_2$O, incubate at 37°C for 1 hr, 96°C for 6 min, then 0.1 °C/s temperature reduction to 23°C), then Golden Gate cloning (1 µl annealed oligo, 75 ng pHCas9, 2 µl T4 DNA Ligase Buffer, 1 µl Golden Gate Enzyme Mix, dH$_2$O to 20 µl), incubation at 37°C for 1 hr then 60°C for 5 min, followed by direct transformation of 1 µl, plated on Lb with spectinomycin. Note that pHCas9-3 was the most inhibitory (**Figure 1—figure supplement 2A**) and was the only plasmid used in accumulation during ORACLE. This assembly was also not tolerated in higher copy number plasmids. All plasmid backbones and gene fragments are lab stocks.

## General bacteria and phage methods

Bacterial concentrations were determined by serial dilution of bacterial culture (1:10 or 1:100 dilutions made to 1 ml in 1.5 microcentrifuge tubes in Lb) and subsequent plating and bead spreading of 100 µl of a countable dilution (targeting 50 colony-forming units) on Lb plates. Plates were incubated overnight and counted the next morning. Typically, 2–3 dilution series were performed for each host to initially establish concentration at different OD$_{600}$ and subsequent concentrations were confirmed with a single dilution series for later experiments.

Stationary phase cultures are created by growing bacteria overnight (totaling ~20–30 hr of incubation) at 37°C. Cultures are briefly vortexed, then used directly. Exponential phase culture consists of stationary culture diluted 1:20 in Lb, then incubated at 37°C until an OD$_{600}$ of ~0.4–0.8 is reached (as determined using an Agilent Cary 60 UV-Vis Spectrometer using manufacturer's documentation),

typically taking 40 min to 1 hr and 20 min depending on the strain and antibiotic, after which cultures are briefly vortexed and used directly.

Phage lysate was purified by centrifuging phage lysate at 16 $g$, then filtering supernatant through a 0.22 µM filter. Chloroform was not routinely used unless destruction of any remaining host was considered necessary and is mentioned in such cases.

To establish titer, phage samples were typically serially diluted (1:10 or 1:100 dilutions made to 1 ml in 1.5 microcentrifuge tubes) in Lb to a $10^{-8}$ dilution for preliminary titering by spot assay. Spot assays were performed by mixing 250 µl of relevant bacterial host in stationary phase with 3.5 ml of 0.5% top agar, briefly vortexing, then plating on Lb plates warmed to 37℃. After plates solidified (typically ~5 min), 1.5 µl of each dilution of phage sample was spotted in series on the plate. Plates were incubated and checked at 2–4 hr and in some cases overnight (~20–30 hr) to establish a preliminary titer. After a preliminary titer was established, phage samples were serially diluted in triplicate for EOP assays. EOP assays were performed using whole plates instead of spot plates to avoid inaccurate interpretation of results due to spotting error (*Khan Mirzaei and Nilsson, 2015*). To perform the whole plate EOP assay, 250 µl of bacterial host in stationary or exponential phase was mixed with between 5 and 50 µl of phages from a relevant dilution targeted to obtain 50 plaque-forming units (PFUs) after overnight incubation. The phage and host mixture was briefly vortexed, briefly centrifuged, then added to 3.5 ml of 0.5% top agar, which was again briefly vortexed and immediately plated on Lb plates warmed to 37℃. After plates solidified (typically ~5 min), plates were inverted and incubated overnight. PFUs were typically counted at 4–6 hr and after overnight incubation (~20–30 hr), and the total overnight PFU count used to establish titer of the phage sample. PFU totals between 10 and 300 PFU were typically considered acceptable, otherwise plating was repeated for the same dilution series. This was repeated in triplicate for each phage sample on each relevant host to establish phage titer.

EOP was determined using a reference host, typically *E. coli* 10G with pHT7Helper1 but stated if otherwise. EOP values were generated for each of the three dilutions by taking the phage titer on the test host divided by the phage titer on the reference host, and this value was subsequently $\log_{10}$ transformed. Values are reported as mean ± SD.

MOI was calculated by dividing phage titer by bacterial concentration. MOI for the T7 variant library after the variant gene is expressed was estimated by titering on 10G with pHT7Helper1.

Limit of detection (LOD) for T7 acceptor phages (T7 Acc) and T7 lacking a tail fiber (T7Δ*gp17*) was established based on the ability of these phages to clear a bacterial lawn. These phages are unable to plaque on host lacking pHT7Helper1, but phages do express a tail fiber due to being propagated on host with pHT7Helper1. Functionally this allows these phages to complete one infection cycle and kill one host but does not allow the creation of plaque-viable progeny phages if that host does not also contain pHT7Helper1. At an MOI of greater than ~2, we noted plates no longer form lawns of bacteria but instead contain individual colonies or are clear, reflective of these singular assassinations. As expected, this effect occurs at different concentrations of phages for exponential or stationary host due to different host concentration at those stages of growth. As plaques cannot form under these conditions and these infections are not productive beyond a single infection, we simply used this cutoff as the LOD for this assay.

Growth time courses for UTI473 (*Figure 5*) and OD$_{600}$ were performed using a Synergy HTX Multi-Mode 96-well plate reader, using 140 µl of host and 10 µl of relevant phage titers. Phages were applied after an hour of incubation in the plate reader.

## Recombination rate and accumulation assays

To establish recombination rate (*Figure 1C*), we passaged T7 acceptor phages on 5 ml exponential phase *E. coli* 10G containing pHT7Helper1 and pHRec1. pHRec1 was used because the recombined *gp17* tail fiber is wildtype, ensuring every recombined phage is plaque-capable (derived from *Figure 1—figure supplement 2B* results). We sought to evaluate recombination rate after only one passage through the host to avoid misinterpretation of results in case recombined phages had different fitness than unrecombined phages. We used an MOI of 10 and allowed passage for 30 min, sufficient time for one wildtype phage passage, after which we halted any remaining reactions by adding 200 µl of chloroform and lysing the remaining bacterial host. Phages were then purified to acquire the final phage population. We established the phage population titer on 10G and 10G with pHT7Helper1. Both acceptor phages and recombined phages are capable of plaquing on 10G with

pHT7Helper1, and this phage titer is used to count the total phage population. Only recombined phages are capable of plaquing on 10G, and this titer is used to count recombined phages. Recombination rate was established as the fraction titer of recombined phages divided by recombined phages. This was repeated in triplicate and reported as mean ± SD.

## Method note

It should be noted that this assay does not delineate for when recombination occurs in the host or how frequently recombination occurs in any one host. For example, if recombination were to occur on the original phage genome, all subsequent progeny phages could contain the recombined gene. In contrast, if recombination were to occur on the phage genome while it is being replicated, anywhere from one individual progeny to all progeny could contain the recombined gene.

To validate accumulation of recombined phages over acceptor phages (*Figure 1D, Figure 1—figure supplement 2E, F*), we first generated a population of recombined phages using the same scheme as outlined for the recombination rate assay. After recombination, this phage population contained primarily T7 acceptor phages with a small percentage of recombined phage containing a wildtype *gp17* tail fiber. This phage population was passaged on 10G containing pHT7Helper1 and either pHCas9-3 (targeting the fixed region in the acceptor phage using g3, the most effective guide by EOP, *Figure 1—figure supplement 2A*) or pHCas9 (randomized control). Phages were incubated with host in 5 ml total at an initial MOI of 1 based on the titer of the whole phage population. Every 30 min until 180 min, and thereafter every 60 min until 300 min, ~250 µl of culture was removed, infection was stopped by adding 100 µl of chloroform, and phage samples were purified to establish the phage population at that timepoint. Titer at each timepoint was determined on both 10G and 10G with pHT7Helper1 with a single dilution series using whole-plate plaque assay. Percent accumulation was derived by dividing titer on 10G by titer on 10G with pHT7Helper1. Accumulation on both hosts was repeated in triplicate and reported as mean ± SD.

## DMS plasmid library preparation

To create the DMS variant plasmid library, oligos were first designed and ordered from Agilent as a SurePrint Oligonucleotide Library (product G7220A, OLS 131-150mers). Every oligo contained a single substitution at a single position in the tip domain, overall including all non-synonymous substitutions, a single synonymous substitution, and a stop codon from position 472–554. Note that we did mutate the stop codon, which is position 554, that when substituted results in a three amino acid extension (-DAR) of *gp17*. We used the most frequently found codon for each amino acid in the *gp17* tail fiber to define the codon for each substitution. Oligos contained BsaI sites at each end to facilitate Golden Gate cloning. To accommodate a shorter oligo length, the library was split into three pools covering the whole tip domain. Oligo pools were amplified by PCR using 0.005 pmol total oligo pool as template and 15 total cycles to prevent PCR bias, then pools were purified. pHRec1 was used as template in a PCR reaction to create three backbones for each of the three pools. Backbones were treated with BsaI and Antarctic Phosphatase as follows. A 5 µl 10× CutSmart, 2 µl BsaI, ~1177 ng backbone, dH$_2$O to 50 µl was mixed and incubated at 37˚C for 2 hr, after which 1 µl additional BsaI, 2 µl Antarctic Phosphatase, 5.89 µl 10× Antarctic Phosphatase buffer was spiked into reaction. Reaction was incubated for one more hour at 37˚C, then enzymes were heat inactivated at 65˚C for 20 min (concentration ~20 ng/µl at this point) and used directly (no purification) in Golden Gate Assembly. Golden Gate Assembly was performed using ~100 ng of relevant pool backbone and a 2× molar ratio for oligos (~10 ng), combined with 2 µl 10× T4 DNA ligase buffer, 1 µl NEB Golden Gate Enzyme Mix, and dH$_2$O to 20 µl. These reactions were cycled from 37˚C to 16˚C over 5 min, 30×, then held at 60˚C for 5 min to complete Golden Gate Assembly. Membrane drop dialysis was then performed on each library pool for 75 min to enhance transformation efficiency. Then, 2 µl of each pool was transformed into 33 µl competent *E. coli* 10-beta (NEB C3020) cells. Drop plates were made at this point (spotting 2.5 µl of dilutions of each library on Lb plates with spectinomycin), and total actual transformed cells were estimated at ~2 × 10$^5$ CFU/ml. Each 1 ml pool was added to 4 ml Lb with spectinomycin and incubated overnight, then plasmids were purified. Plasmids concentration was determined by nanodrop and pools were then combined at an equimolar ratio to create the final phage variant pool, denoted as pHRec1-Lib. pHRec1-Lib was transformed into *E. coli* 10G with pHT7Helper1. Drop plates were made (spotting 2.5 µl of dilutions

of each library onto Lb plates with spectinomycin and kanamycin) and total actual transformed cells were also estimated at ~2 × 10^5 CFU/ml. The 1 ml library was added to 4 ml Lb with spectinomycin and kanamycin and incubated overnight. This host, *E. coli* 10G with pHT7Helper1 and pHRec1-Lib, was the host used for Optimized Recombination during ORACLE.

## ORACLE: engineering T7 acceptor phages

Acceptor phages were assembled using YAC rebooting (*Ando et al., 2015*; *Jaschke et al., 2012*), which requires yeast transformation of relevant DNA segments, created as follows. A prs415 yeast centromere plasmid was split into three segments by PCR, separating the centromere and leucine selection marker, which partially limits recircularization and improved assembly efficiency (*Kuijpers et al., 2013*). Wildtype T7 segments were made by PCR using wildtype T7 as template. At the site of recombination, the acceptor phage contains, in order, lox71 sites with an m2 spacer (*Langer et al., 2002*) to facilitate one-way recombinase-mediated cassette exchange (RMCE), a fixed sequence that was derived from sfGFP with a nonsense mutation, a short region mimicking *gp17* to allow detection of acceptor phages by deep sequencing (5′ NGS in *Figure 1—figure supplement 1*), a 3′ 'pad' to facilitate deep sequencing, and lox66 sites with a wt spacer (see *Figure 1—figure supplement 1*). This entire region was turned into one DNA segment by serial SOE reactions.

Method note: Reversible or two-way RMCE using wildtype loxP sites could feasibly increase recombination efficiency as one-way RMCE is not necessarily required for ORACLE.

Method note: PCR, including PCR for deep sequencing, behaved inconsistently at lox sites. We theorize that this may be because these sites are inverted repeats. Our inclusion of the 3′ 'pad' region corrected this problem and facilitated acceptor phage detection by deep sequencing.

DNA parts were combined together (0.1 pmol/segment) and transformed into *S. cerevisiae* BY4741 using a high-efficiency yeast transformation protocol (*Gietz and Woods, 2002*) using SD-Leu selection. After 2–3 days, colonies were picked and directly assayed by multiplex colony PCR to assay assembly. Multiplex PCR interrogated junctions in the YAC construct and was an effective way of distinguishing correctly assembled YACs. Correctly assembled YACs were purified and transformed into *E. coli* 10G cells containing pHT7Helper1, and after recovery 400 µl was used to inoculate 4.6 ml Lb. This culture was incubated until lysis, after which phages were purified to create the acceptor phage stock.

## ORACLE: Optimized Recombination

Recombination was performed by adding T7 acceptor phages (MOI ~ 5) to 15 ml exponential phase 10G with pHT7Helper1 and pHRec1-Lib (shown as the donor plasmid in *Figure 1*), split across three 5 ml cultures. A high MOI is used to allow for one effective infection cycle. Cultures were incubated until lysis (~30 min). Lysed cultures were combined and purified. This phage population constitutes the initial recombined phage population and contained an estimated 2 × 10^7 variants/ml in a total phage population of ~2 × 10^10 PFU/ml. The remainder of the phages are acceptor phages. A schematic of the recombination is shown in *Figure 1—figure supplement 1*. Note that pHT7Helper1 ensures progeny should remain viable by providing *gp17 in trans*.

## ORACLE: Accumulation

Accumulation was performed by adding ~MOI of 0.2 of recombined phages (50 µl or ~1 × 10^9 total phages) to 5 ml of stationary phase *E. coli* 10G with pHT7Helper1 and pHCas9-3 resuspended in fresh Lb with kanamycin and spectinomycin. Cultures are incubated until lysis (~3.5 hr), then phages are purified. This MOI was chosen to target 1% of acceptor phages remaining in the final population as an internal control – the remainder of the phage population is accumulated variant phages. Stationary phase was used because it was more inhibitory based on EOP (*Figure 1—figure supplement 2A*). Note that pHT7Helper1 still ensures progeny should remain viable by providing *gp17 in trans* and progeny from accumulation do not fully express variant genes.

Method note: During ORACLE, the library gene is not actively repressed and some fraction of progeny phages are likely assembled with the variant gene or contain chimeric tail fibers with both proteins. This may account for some degree of the skew we see during accumulation, although skewed residues were not consistent with enrichment patterns on 10G. Here, this skew was not significant, but this could be an influencing factor in future studies. The bias can feasibly be further

reduced by a number of methods of varying complexity, including repressing the genomic variant, increasing the amount of wildtype provided in trans, or simply reducing the number of replication events during ORACLE.

Method note: See *Figure 1—figure supplement 2A* for inhibition results for versions of pHCas9. When sequenced, individual plaques after selection had mutations in the region targeted by each gRNA, as expected for how resistance to Cas9 occurs. Note that acceptor phages are not actively destroyed but are rather inhibited and maintained at the same concentration. Selection may be improved by using multiple guide RNAs or using sgRNA platforms.

## ORACLE: Library Expression

Library expression was performed by adding the accumulated DMS library to 5 ml *E. coli* 10G (with no plasmid) at an MOI of ~1. Cultures are incubated for 30 min, then 200 μl chloroform is added to the culture to lyse any remaining cells and phages are purified. This constitutes the final phage variant library with full expression of the variant *gp17* tail fibers. This phage population is directly sequenced to establish the pre-selection library population.

Method note: This MOI and culture conditions are chosen to prevent phages from undergoing more than one replication cycle. Additional replication cycles would result in skew toward variants that grow better on the host used for expression. At an MOI of 1, progeny from the first replication cycle already comprise the majority of the possible concentration of phages for T7. Chloroform is added at 30 min to halt any rogue second infections in process.

Method note: Two points bear additional mention regarding ORACLE as a whole to create variant libraries. First, the importance of retaining variants that do not grow well on the host used to create the library cannot be overstated. These variants are critical for mapping functional regions. For example, we used 10G to grow our library, which happened to have the most significant selection of the susceptible hosts and had depletion of many functional regions. The resolution of this assay would have been deeply impacting if these variants had been lost. Second, due to possible depletion and skew, it is critical to assay library distribution after insertion and expression of the variants in the phage instead of, for example, assaying distribution in plasmid before phage insertion. While ORACLE is designed to avoid this problem, any selection that occurs during library construction needs to be identified prior to selection experiments.

## DMS selection

All DMS selection results, besides single-infection cycle experiments on resistant hosts noted later, were performed in the same way. The T7 variant library was added to 5 ml of exponential host at an MOI of ~$10^{-2}$, and the culture was allowed to fully lyse (typically 40–80 min depending on the host). Phage lysate was purified and then the titer established for the host the phage was being selected on. This process was then repeated using the selected phage lysate. An MOI of $10^{-2}$ was chosen to allow phages that grow slower a chance to replicate. For reference under these conditions, we expect wildtype to complete four infection cycles on a susceptible host. Phage lysate from the second selection was retained and used as template for deep sequencing to establish the post-selection phage population. The entire process was repeated in biological triplicate for each host. Single-infection cycle results cited in *Figure 4—figure supplement 4* and *Figure 4—source data 1* and *2* were performed on BW25113Δ*rfaG* and BW25113Δ*rfaD* using an MOI of ~1, then waiting for the culture to fully lyse, after which phage lysate was purified and sequenced directly.

Method note: The number of infection cycles should be carefully considered due to the effect of multiple exponential replication events on the phage population. As noted in the text, we expected less consistency as we increased infection cycles, especially on severely bottlenecked populations as seen on BW25113Δ*rfaD*, but additional infection cycles allow for increased competition between productive members.

## Deep sequencing preparation and analysis

We used deep sequencing to evaluate phage populations. We first amplified the tip domain by two-step PCR, or tailed amplicon sequencing, using KAPA HiFi. Primers for deep sequencing attach to constant regions adjacent to the tip domain (the target region is 304 bp total, between the 5′ NGS region and 3′ pad on *Figure 1—figure supplement 1*). Constant regions are also installed in the

fixed region of the acceptor phages for the same-size amplicon, so acceptor phages can also be detected. The first PCR reaction adds an internal barcode (used for technical replicates to assay PCR skew), a variable N region (to assist with nucleotide diversity during deep sequencing, this is essential for DMS libraries due to low nucleotide diversity at each position), and the universal Illumina adapter. Undiluted phages are used as template. Four forward and four reverse primers were used in each reaction, each with a variable N count (0, 2, 4, or 8). Primers were mixed at equimolar ratios, and total primers used was per recommended primer concentration. PCR was performed using 12 total cycles in the first PCR reaction, then the product of this reaction was purified. The second PCR reaction adds an index and the Illumina 'stem'. Then, 1 μl of purified product from the first reaction was used as template using eight total PCR cycles. The product of this reaction was purified and used directly for deep sequencing. Each phage population was sampled at least twice using separate internal barcodes, and no PCR reactions were pooled. Total PCR cycles overall for each sample were kept at 20× to avoid PCR skew. All phage samples were deep sequenced using an Illumina MiSeq System, 2 × 250 read length using MiSeq Reagent Kit v2 or v2 Nano according to manufacturer's documentation.

Paired-end Illumina sequencing reads were merged with FLASH (Fast Length Adjustment of SHort reads) using the default software parameters (*Edgar and Flyvbjerg, 2015*). Phred quality scores (Q scores) were used to compute the total number of expected errors (E) for each merged read, and reads exceeding an Emax of 1 were removed (*Magoč and Salzberg, 2011*). Wildtype, acceptor phages, and each variant were then counted in the deep sequencing output. We correlated read counts for each technical replicate to determine if there was any notable skew from PCR or deep sequencing. Replicates correlated extremely well (R ≥ 0.98 for all samples), indicating no relevant PCR skew. Besides wildtype and acceptor counts, we included only sequences with single substitutions in our analysis. While this limited the scope of the analysis, it greatly reduced the possibility of deep sequencing error, resulting in an incorrect read count for a variant, as virtually every relevant error would result in at least a double substitution in our library. With this in mind and to avoid missing low abundance members after selection, we simply used a low read cutoff of 2 and did not utilize a pseudo-count of 1 for each position.

Of the 1660 variants, 3 (S487P, L524M, and R542N) fell below our LOD in the variant pool before selection. These positions were excluded from analysis as a pre-selection population could not be accurately determined, although both S487P and L542M subsequently emerged in several post-selection populations, indicating that they were present below the LOD. Technical replicates of each biological replicate were aggregated and each biological replicate was correlated to determine reproducibility, reported for each relevant figure as a figure supplement. Possible outliers were identified by data sorting, and lower correlation in 10G was noted to primarily be a result of differences in enrichment of several polar, uncharged substitutions and proline substitutions in the first biological replicate. These substitutions are proximally located near exterior loop HI and may indicate an unknown variable or growth condition in the first replicate that produces a slightly different enrichment pattern. Correlation was otherwise robust, and excluding only G479Q produces R = 0.94, 0.95, and 0.99 as displayed in *Figure 2—figure supplement 1*. Positively charged, downward-facing substitutions were universally enriched in BW25113Δ*rfaD*, but correlation was reduced due to inconsistencies in which particular substitutions were the most enriched in a given replicate. While the same substitutions were enriched in all three replicates, suggesting reproducibility of results, the scale of enrichment varied considerably. Correlation was otherwise robust, and excluding the single most enriched substitution of each replicate (G521R, A500R, and G521H) produces R = 0.90, 0.93, and 0.86 as displayed.

Method note: BW25113Δ*rfaD* is the most resistant host by EOP, and we hypothesize that these inconsistencies in $F_N$ may arise due to severe loss of diversity, 'bottlenecking' the population, and causing stochastic differences in enrichment to become magnified with multiple rounds of selection across independent experiments. Future work with very resistant hosts may benefit from fewer rounds of selection to prevent significant stochastic divergence. Single-infection cycles are more consistent but provide fewer cycles for variants to directly compete with one another, as noted in methods for DMS selection.

To score enrichment for each variant, we used a basic functional score (F), averaging results of the three biological replicates, where $F = \bar{x}\, \frac{Variant\,\%_{Post-Passage}}{Variant\,\%_{Pre-Passage}}$. To compare variant performance across hosts, we normalized functional score ($F_N$) to wildtype, where $F_N = \bar{x}\, \frac{Variant\,\%_{Post-Passage}}{Variant\,\%_{Pre-Passage}} \big/ \frac{WT\,\%_{Post-Passage}}{WT\,\%_{Pre-Passage}}$.

## Classifying variants and isolating variants

To define variant behavior on *E. coli* 10G, BL21, and BW25113, we considered variants depleted if $F_N$ was below 0.1 (i.e., performed 10 times worse than wildtype), tolerated if between 0.1 and 2, and enriched if above 2 (i.e., performed twice as good as wildtype) (*Figure 2*). As wildtype T7 effectively grows on all three hosts, we reasoned that it would be more challenging for an enriched mutant to surpass wildtype than it would be for a mutant to become depleted. These cutoffs were supported based on preliminary plaque assay results and the extent of standard deviation across biological replicates. For BW25113Δ*rfaD* and BW25113Δ*rfaG*, we further defined significantly enriched variants as performing at least 10 times better ($F_N \geq 10$) than wildtype because wildtype does not grow effectively on these strains (*Figure 4*).

We compared variant $F_N$ across 10G, BL21, and BW25113 to further characterize each variant and find functional variants (*Figure 3*). We sought to identify variants that had meaningfully different performance on different hosts, which would be strong evidence that either the wildtype residue or the variant substitution was important in a host-specific context. In addition to providing direct insight intro structure–function relationships, such substitutions or positions are ideal engineering targets for altering host range or increasing activity in engineered phages. We considered substitutions that were depleted ($F_N < 0.1$) on all three hosts to be intolerant, while substitutions that were tolerated or enriched ($F_N \geq 0.1$) in all three hosts to be generally tolerated. Substitutions that were depleted on one host but tolerated or enriched on another were considered functional. To broadly characterize each position, we counted the number of substitutions at that position that fell into each category, and colored positions (*Figure 3C*) as functional if over 33% of substitutions at that position were functional, intolerant if over 50% of substitutions were intolerant, and tolerant otherwise. We found these cutoffs to effectively group residues of interest, although we note that there remain substitutions that could be tolerant and relevant in different contexts or intolerant for these three hosts but not others.

For defining ideal host constriction mutants (*Figure 6*), we first constricted $F_N$ values that were greater than 1 to reduce the impact of higher scores on this comparison. Specifically we generated functional difference ($F_D$), where if $F_N < 1$, $F_D = F_N$, and where $F_N \geq 1$, $F_D = \frac{F_N - 1}{\max(Strain\ F_N) - 1} + 1$. $F_D$ thus ranged from 0 to 2 for each substitution, where scores above 1 are normalized to the maximum value for that host and fall between 1 and 2, minimizing but not eliminating weight for enrichment. We reasoned that for the purposes of finding host constriction mutants the extent of enrichment for a substitution is less relevant than if that substitution did poorly on another host. Put another way, it does not matter if a substitution is tolerated or enriched so long as it is depleted on a different host. For example, V544R has an $F_N$ of 9.09 in *E. coli* 10G but 0.07 in *E. coli* BW25113, while G479E has an $F_N$ of 1.73 in *E. coli* 10G and falls below the LOD for *E. coli* BW25113. For host constriction, both positions should be scored highly as the mutations can be tolerated or enriched in one host but are depleted in another. In contrast, A500H has an $F_N$ of 7.46 in *E. coli* 10G and 1.2 in *E. coli* BW25113. While $F_N$ differs significantly and the substitution is enriched on one host, it is still tolerated in the other host and thus makes a poor host constriction target. After generating $F_D$, we simply subtracted the substitution's $F_D$ on one host from the other in a pairwise comparison (*Figure 6*). Substitutions for host range constriction were considered ideal candidates if $|F_D| \geq 1$.

Variants with individual substitutions tested for EOP (*Figures 4–6*) were either picked from plaques or created using SDM on pHRec1, which was used to create the variant using ORACLE.

## Rosetta ΔΔG and physicochemical property comparison and calculations

The crystal structure of the *gp17* tip domain was obtained (PDB ID: 4A0T) and water molecules removed before calculations run. All modeling calculations were performed using the Rosetta molecular modeling suite v3.9. Substitutions were generated using the standard ddg_monomer application (*Kellogg et al., 2011*) to enable local conformational to minimize energy. For comparison to $F_D$ (Figure S5), a ΔΔG of 10 or greater was considered destabilizing and ΔΔG values between 10 and 30

were transformed to values between 0 and 1, with any $\Delta\Delta G$ greater than 30 set to 1. $\Delta\Delta G$ values below 10 were transformed to values between 0 and $-1$ based on a range to $-9.29$, the most stabilizing $\Delta\Delta G$ value calculated. We calculated $F_D$ for this plot using the maximum $F_D$ value from *E. coli* 10G, BL21, or BW25113. The maximum was used because any substitution that has a high $F_D$ on any host was theorized to require a stable protein. Positions considered destabilizing (right side) are expected to have a very low maximum $F_D$, whereas stabilizing positions (left side) may have a low or high $F_D$ based on tolerance of the substitution. Positions that were tolerated or functional that Rosetta predicted to be unstable on the right of the plot may be due to errors in stability calculations or actual structural distortions that are either smaller perturbations that do not affect fitness or are accommodated by quaternary arrangement. Alternatively, structural instability could be beneficial in some cases, allowing for enhanced receptor binding by, for example, exposing critical residues.

To compare physicochemical properties for *E. coli* 10G, BL21, and BW25113, we binned depleted ($F_N \leq 0.1$), tolerated ($F_N > 0.1$ and $<2$), or enriched ($F_N \geq 2$) substitutions and derived the change in mass, hydrophilicity, and hydrophobicity for each substitution (*Li et al., 2016*). For BW25113$\Delta rfaG$ and BW25113$\Delta rfaD$, we binned using an $F_N$ of 10 for the cutoff for enrichment instead. Packages ggplot2 and ggpubr in R were then used to compare the means of the three groups using a Kruskal–Wallis test (*Hollander and Wolfe, 1973*), while subsequent pairwise comparisons were made using a Wilcoxon test (*Bauer, 1972*). Effect size (r) was calculated and can be summarized as r = abs (zscore)/sqrt(n), where n is the total number of observations among the two groups.

## Statistical analysis and source code

Alluvial plots (*Figure 2—figure supplement 2*, *Figure 4*) and *Figure 2F* parallel plot were generated with RawGraphs. Violin plots for physicochemical properties are output from R. Significance for $F_N$ values in *Figure 2—source data 2* and *Figure 4—source data 2* was defined as an average $F_N$ two or more standard deviations from wildtype or below the LOD in all biological replicates. p-Values for EOP graphs compare plaque capability on the tested host to the reference host for the EOP graph. Hierarchical clustering in *Figure 2—figure supplement 3* is performed in R using heatmap.2 without scaling. p-Values to compare functional data and EOP measurements were performed using two-tailed *t*-tests where values below the LOD were considered as the LOD. All other calculations and plots were made in Excel. Relevant statistical details of experiments can be found in the corresponding figure legends or relevant methods section. Source code is available at https://github.com/raman-lab/oracle.

## Acknowledgements

We thank Dr. Rodney Welch for UTI473 strain and Dr. Douglas Weibel for BW25113 deletion mutant strains. We thank Dr. Karthik Anantharaman, Dr. John Yin, Laura Alexander, and Chutikarn Chitboonthavisuk for critical review of the manuscript. This work was partially supported by the US Department of Agriculture Hatch award (WIS02066) and the Gates Grand Challenges grant (OPP1150209). AM is supported by the Great Lakes Bioenergy Research Center (U.S. Department of Energy Award Number DE-SC0018409). ML is supported by NIH Molecular Biophysics Training Program T32 GM08293 and William H Peterson Fellowship in Biochemistry. KN is supported by NIH National Research Service Award T32 GM07215 and the Robert and Katherine Burris Biochemistry Fund.

## Additional information

### Competing interests

Phil Huss: PH and SR have filed a provisional patent application on this technology (patent application number WIS0055US). Srivatsan Raman: PH and SR have filed a provisional patent application on this technology (patent application number WIS0055US). SR is on the scientific advisory board of MAP/PATH LLC. The other authors declare that no competing interests exist.

## Funding

| Funder | Grant reference number | Author |
|---|---|---|
| U.S. Department of Agriculture | WIS02066 | Srivatsan Raman |
| Bill and Melinda Gates Foundation | OPP1150209 | Srivatsan Raman |
| U.S. Department of Energy | DE-SC0018409 | Anthony Meger |
| NIH | T32 GM08293 | Megan Leander |
| NIH | T32 GM07215 | Kyle Nishikawa |
| Robert and Katherine Burris | BiochemistryFund | Kyle Nishikawa |

The funders had no role in study design, data collection and interpretation, or the decision to submit the work for publication.

## Author contributions

Phil Huss, Conceptualization, Data curation, Software, Formal analysis, Investigation, Visualization, Methodology, Writing - original draft, Writing - review and editing; Anthony Meger, Software; Megan Leander, Kyle Nishikawa, Methodology; Srivatsan Raman, Conceptualization, Resources, Supervision, Funding acquisition, Project administration, Writing - review and editing

## Author ORCIDs

Phil Huss (iD) https://orcid.org/0000-0003-4064-9333
Srivatsan Raman (iD) https://orcid.org/0000-0003-2461-1589

## Decision letter and Author response

Decision letter https://doi.org/10.7554/eLife.63775.sa1
Author response https://doi.org/10.7554/eLife.63775.sa2

# Additional files

## Supplementary files

- Supplementary file 1. List of primers used for experimentation.
- Transparent reporting form

## Data availability

Source code has been deposited on github here: https://github.com/raman-lab/oracle (copy archived at https://archive.softwareheritage.org/swh:1:rev:657e8eef12e4ee886f5d188b745ff0b38f94f479/). Raw NGS data is publicly available via Zenodo (doi: 10.5281/zenodo.4583797). Other data generated or analyzed during this study are included in the manuscript and supporting files. Source data files have been provided for Figures 1, 2, 3, 4 and 6.

The following dataset was generated:

| Author(s) | Year | Dataset title | Dataset URL | Database and Identifier |
|---|---|---|---|---|
| Huss P, Meger A, Leander M, Nishikawa K, Raman S | 2021 | Raw NGS Data for "Mapping the functional landscape of the receptor binding domain of T7 bacteriophage by deep mutational scanning" | http://doi.org/10.5281/zenodo.4583797 | Zenodo, 10.5281/zenodo.4583797 |

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
