## [Decision Letter]

**Acceptance summary:**

Huss et al. have developed a novel tool (ORACLE) for generating libraries of phage variants. They go on to apply this tool to study the residues important for T7 host specificity, providing a rich dataset for in-depth functional studies. They validate a subset of hits and use this information to engineer T7 variants that may be able to overcome bacterial resistance against a urinary tract infection associated strain, consistent with their in vitro results. Their approach provides both a valuable new tool and intriguing biological insights prompting future studies.

**Decision letter after peer review:**

Thank you for submitting your article "Mapping the Functional Landscape of the Receptor Binding Domain of T7 Bacteriophage by Deep Mutational Scanning" for consideration by *eLife*. Your article has been reviewed by four peer reviewers, one of whom is a member of our Board of Reviewing Editors, and the evaluation has been overseen by Gisela Storz as the Senior Editor. The following individuals involved in review of your submission have agreed to reveal their identity: Laurent Debarbieux (Reviewer #2); Breck A Duerkop (Reviewer #3); James S Fraser (Reviewer #4).

The reviewers have discussed the reviews with one another and the Reviewing Editor has drafted this decision to help you prepare a revised submission.

Summary:

Huss et al. describe a phage genome engineering technology that they call ORACLE. This technique uses recombineering of a phage target gene with a variant library to identify both gain and loss of function mutations. The beauty of this method and what makes it superior to other techniques is that it dramatically limits loss of mutants that are less fit during the initial round of library generation. Thus, the pool of variants is vast and is reduced in bias toward more fit species based on the host used for initial library amplification. They use the model coliphage T7 as a proof of principal and show that several previously unidentified residues of in the T7 tail fiber play critical roles in both loss and gain of function for phage infectivity and they also identify residues that are major drivers of altered host tropism. Lastly, they apply this library to a pathogenic UTI associated strain of *E. coli* which is normally resistant to wild type T7 infection and identify tail variants of T7 that can now infect this strain, highlighting the applicability of this method toward the discovery of engineered phages that could be used therapeutically. Altogether this is an important advancement in phage engineering that shows potential promise for future phage therapies.

Essential revisions:

1) Claims about generalizability should either be removed, qualified with the various caveats, or supported by additional data. This study focused on a single phage gene and a single host bacterial species. As such, it is not clear if ORACLE will work well in other contexts. More concerningly, the lack of reproducibility across technical replicates in some of the experiments (e.g., subsection “Discovery of gain-of-function variants against resistant hosts”) may indicate that this method will not work for other T7 genes or phenotypes of interests.

The authors state that ORACLE overcomes three major hurdles that make it better than existing methods, one of which is "generalizability for virtually any phage", while denouncing other systems for being applicable for highly transformable hosts only. This is highly exaggerated since ORACLE requires transformation of two plasmids (helper and donor) including one with tunable gene expression, which is clearly not possible in many bacteria. Furthermore, the enrichment step requires a strain with a functional CRISPR/Cas9 system, which again is not so obvious in the bacterial world.

T7 and its *E. coli* hosts are domesticated strains where phage engineering is considered easier than less well studied phages and their hosts. Considering the authors indicate that the ORACLE method could be applied to any phage-bacteria pair, I would like to see just how feasible it is to generate a highly diverse library on a phage-host pair that are not as well studied as T7-*E. coli*. This is the situation that would likely occur therapeutically.

2) The description and reasoning behind the use of the helper plasmid carrying the wild type tail fiber is not clear as described. This is really what reduces the bias in the first round of library generation and is critical to the technology. I had to re-read this section several times to fully understand the purpose of this. It would be nice to illustrate this in more detail in Figure 1A, showing that the first round of phage packaging of variants is in to particles that most likely have WT tail fibers, thus all phages generated regardless of the variant DNA packaged should in theory have an equal chance of infecting a host and being propagated in the accumulation stage.

---

## [Author Response]

Essential revisions:1) Claims about generalizability should either be removed, qualified with the various caveats, or supported by additional data. This study focused on a single phage gene and a single host bacterial species. As such, it is not clear if ORACLE will work well in other contexts.

In the revision, we more clearly state that a caveat of ORACLE is that the propagating host be transformable and be capable of maintaining a plasmid library (see changes to manuscript below). While this requirement limits the broader applicability of ORACLE, many species of basic science and therapeutic relevance such as *Acinetobacter baumannii, Neisseria gonorrhoeae, Campylobacter jejuni, Enterococcus faecalis, Pseudomonas aeruginosa, Salmonella typhimurium and Salmonella typhi* are sufficiently transformable to be amenable to ORACLE. Additionally, techniques like conjugation could be leveraged to establish a plasmid library in more refractory strains such as *Clostridium difficile*.

The reviewer is correct that ORACLE may be difficult to implement for an arbitrary phage-bacterial system, and we have made modifications to our claims in the revision as follows.

Changes to manuscript:

“Our approach presents a generalized framework for characterizing sequence function relationships in many phage-bacterial systems.”

“ORACLE allows sequence programmability and generalizability to phages with transformable bacterial hosts capable of maintaining a plasmid library.”

“Any phage, including lysogenic phages, with a sequenced genome and a transformable host that can maintain a plasmid library should be amenable to ORACLE.”

“ORACLE is designed as a foundational technology to elucidate sequence-function relationships in phage genes.”

“Any phage with a sequenced genome and a transformable host capable of maintaining a plasmid library should be amenable to ORACLE because the phage variants are created during the natural infection cycle.”

More concerningly, the lack of reproducibility across technical replicates in some of the experiments (e.g., subsection “Discovery of gain-of-function variants against resistant hosts”) may indicate that this method will not work for other T7 genes or phenotypes of interests.

Our data shows high reproducibility across biological replicates in 4 out of 5 deep mutational scanning experiments carried out in this paper (Figure 2—figure supplement 1 and Figure 4—figure supplement 1). However, on one host, BW25113∆*rfaD*, correlation across biological replicates is lower for which our explanation is as follows. T7 variants likely experience the strongest selection pressure on BW25113∆*rfaD* among all hosts tested in this paper, because this strain encodes the largest truncation of the phage receptor (surface LPS) required for T7 infection. This results in a population with low sequence diversity after selection where stochastic differences in experimental conditions across replicates may give rise to greater variability in enrichment due to exponential phage amplification. This effect is increased during the estimated four infection cycles used for selection. On permissive hosts, this effect is buffered by the larger diversity of viable T7 variants leading to greater consistency between replicates.

To address the reviewer’s concern, we have included two additional data items:

1) Ranking of the top T7 variants on BW25113∆*rfaD*: Although the enrichment scores on BW25113∆*rfaD* were variable, the ranking of the top T7 variants across different replicates was consistent, underscoring the robustness of the assay. We have added a new supplementary figure (Figure 4—figure supplement 3) with the top ranked variants on this host and BW25113∆*rfaG*.

2) Correlation plot after one round of selection: We normally compute enrichment scores after four infection cycles on the host which may lead to greater divergence of enrichment scores in a low diversity population. Therefore, we have also added functional scoring data and correlation plots after one round of selection for both BW25113∆*rfaD* and BW25113∆*rfaG* (Figure 4—figure supplement 4 and Figure 4—source data 1 and 2). Data after a single passage is much more consistent (correlation~0.90 on BW25113∆*rfaD*), in line with our hypothesis that additional infection cycles decreases correlation between replicates.

Lower reproducibility on BW25113∆*rfaD* does not diminish the applicability of ORACLE to other T7 genes as suggested by the reviewer. But it does emphasize the importance of experimental design and subsequent interpretation of results.

Changes to manuscript:

We have added Figure 4—figure supplement 3 listing the top ranked variants after selection on BW25113∆*rfaD* and BW25113∆*rfaG* and Figure 4—figure supplement 4 with correlation plots using a single infection cycle for selection. We have updated Figure 4—source data 1 and Figure 4—source data 2 and the Materials and methods appropriately. Specific changes to the main text are as follows:

“Although the scale of F_N_ was inconsistent across replicates on BW25113Δ*rfaD*, the same substitutions were largely enriched in all three replicates, suggesting reproducibility of results (Figure 4—figure supplement 3). […] Separately we examined correlation after selection using only a single infection cycle, which produced more highly correlated results for BW25113Δ*rfaD* (R=0.89, 0.90, 0.89) (Figure 4—figure supplement 4), indicating fewer infection cycles may be ideal for future work with highly resistant hosts.”

“Each variant was given a functional score, F, based on the ratio of their relative abundance before and after selection consisting of an estimated four infection cycles, which was then normalized to wildtype to yield F_N_ where wildtype F_N_ = 1 (Figure 2C-E, see Materials and methods).”

The authors state that ORACLE overcomes three major hurdles that make it better than existing methods, one of which is "generalizability for virtually any phage", while denouncing other systems for being applicable for highly transformable hosts only. This is highly exaggerated since ORACLE requires transformation of two plasmids (helper and donor) including one with tunable gene expression, which is clearly not possible in many bacteria. Furthermore, the enrichment step requires a strain with a functional CRISPR/Cas9 system, which again is not so obvious in the bacterial world.

Our comment regarding highly transformable hosts is in relation to direct transformation of libraries of phage genomes which can be challenging, compared to smaller plasmid libraries in ORACLE. We have modified the sentence as shown below. The reviewer is correct that ORACLE requires maintenance of plasmids and in the revision, we qualify the generalizability of ORACLE with this caveat (also see response to comment 1). However, there is no need for a tunable promoter as the variant library and Cas9 are both expressed from constitutive promoters. It is also worth noting that the role of Cas9 to cleave unrecombined phages can be performed by other site-specific nucleases whose corresponding restriction sequence is inserted in the acceptor phage genome between the recombinase sites.

Changes to manuscript:

“Direct transformation of phage libraries, while ideal for creating one or small groups of synthetic phages, will not work because phage genomes are typically too large for library transformation or are reliant on highly transformable hosts (Ando et al., 2015; Kilcher et al., 2018; Marinelli et al., 2008, 2019).”

T7 and its *E. coli* hosts are domesticated strains where phage engineering is considered easier than less well studied phages and their hosts. Considering the authors indicate that the ORACLE method could be applied to any phage-bacteria pair, I would like to see just how feasible it is to generate a highly diverse library on a phage-host pair that are not as well studied as T7-*E. coli*. This is the situation that would likely occur therapeutically.

We agree with the reviewer that technical hurdles will have to be overcome to extend ORACLE to non-model phage-bacterial systems. However, in our opinion, there is much to learned by simply engineering T7 and other common coliphages against pathogenic *E. coli* strains and related enterobacterial species. Our approach makes this possible because once the phage library is generated on a permissive laboratory host, the variant library can be tested on a several pathogens to identify highly active phage variants. Preliminary results from other ongoing projects in my laboratory suggest efficacy against multiple therapeutically relevant hosts.

2) The description and reasoning behind the use of the helper plasmid carrying the wild type tail fiber is not clear as described. This is really what reduces the bias in the first round of library generation and is critical to the technology. I had to re-read this section several times to fully understand the purpose of this. It would be nice to illustrate this in more detail in Figure 1A, showing that the first round of phage packaging of variants is in to particles that most likely have WT tail fibers, thus all phages generated regardless of the variant DNA packaged should in theory have an equal chance of infecting a host and being propagated in the accumulation stage.

We have altered and simplified Figure 1A to make the role of the helper plasmid clearer in both recombination and accumulation stages and further emphasize the importance of the lack of helper plasmid during expression.

Changes to manuscript:

The graphic in Figure 1A has been simplified and improved to better illustrate the role of the helper plasmid, and the Figure 1 legend has been updated accordingly.

“To minimize biasing of variants during propagation, a helper plasmid constitutively provides the wildtype tail fiber *in trans* such that all progeny phages can amplify comparably regardless of the fitness benefit or deficient of any variant.”

“To enrich recombined phages in this pool, we passage all progeny phages on *E. coli* expressing Cas9 and a gRNA targeting the fixed sequence flanked by recombinase sites we introduced into the acceptor phage. […] As a result, only unrecombined phages will be inhibited while recombined phages with tail fiber variants are Accumulated without bias.”

“In the final step, phages are propagated on *E. coli* which lack the helper plasmid that previously provided the wild type tail fiber *in trans* to prevent bias. In this Library Expression, propagation on this host allows for full expression of the library variant – this is the first time during library creation that the variant is fully expressed on the phage particle.”